# Leveraging Hallucinations to Reduce Manual Prompt Dependency in Promptable Segmentation

**Jian Hu[1], Jiayi Lin[1], Junchi Yan[2], Shaogang Gong[1]**
[1]School of Electronic Engineering and Computer Science, Queen Mary University of London
[2]Dept. of CSE & School of AI & Moe Key Lab of AI, Shanghai Jiao Tong University
{jian.hu, jiayi.lin, s.gong}@qmul.ac.uk, yanjunchi@sjtu.edu.cn
https://lwpyh.github.io/ProMaC/

## Abstract

Promptable segmentation typically requires instance-specific manual prompts to guide the segmentation of each desired object. To minimize such a need, task-generic promptable segmentation has been introduced, which employs a single task-generic prompt to segment various images of different objects in the same task. Current methods use Multimodal Large Language Models (MLLMs) to reason detailed instance-specific prompts from a task-generic prompt for improving segmentation accuracy. The effectiveness of this segmentation heavily depends on the precision of these derived prompts. However, MLLMs often suffer hallucinations during reasoning, resulting in inaccurate prompting. While existing methods focus on eliminating hallucinations to improve a model, we argue that MLLM hallucinations can reveal valuable contextual insights when leveraged correctly, as they represent pre-trained large-scale knowledge beyond individual images. In this paper, we utilize hallucinations to mine task-related information from images and verify its accuracy for enhancing precision of the generated prompts. Specifically, we introduce an iterative **Pro**mpt-**Ma**sk **C**ycle generation framework (ProMaC) with a prompt generator and a mask generator. The prompt generator uses a multi-scale chain of thought prompting, initially exploring hallucinations for extracting extended contextual knowledge on a test image. These hallucinations are then reduced to formulate precise instance-specific prompts, directing the mask generator to produce masks that are consistent with task semantics by mask semantic alignment. The generated masks iteratively induce the prompt generator to focus more on task-relevant image areas and reduce irrelevant hallucinations, resulting jointly in better prompts and masks. Experiments on 5 benchmarks demonstrate the effectiveness of ProMaC. Code given in https://lwpyh.github.io/ProMaC/.

## 1 Introduction

Current promptable segmentation methods rely on instance-specific manual prompts to guide segmentation, greatly limiting its large-scale application. Recently, a manual-free task-generic promptable segmentation approach was introduced [21]: only a single task-generic prompt is needed for all samples under the same task, e.g., "camouflaged animal" is a task-generic prompt for all images in a camouflaged object detection task. The model segments task-relevant objects in various images based on this generic prompt, significantly reducing the annotation workload.

This work was supported by Veritone, Adobe, OpenAI, the Apocrita HPC facility from the QMUL Research-IT, the China Scholarship Council, NSFC (92370201, 62222607) and Shanghai Municipal Science and Technology Major Project under Grant 2021SHZDZX0102. Thanks to Weitong Cai for helpful discussion.

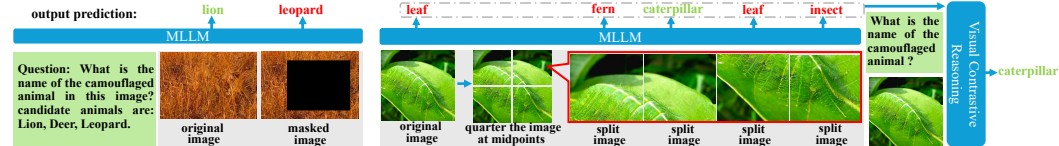

(a) Hallucination by co-occurrence prior.  (b) Using hallucinations can benefit accuracte prompt generation.

Figure 1: (a) During MLLM pretraining, leopards often co-occur with grass. If the lion is masked, the model incorrectly identifies it as a leopard based on the grass. (b) Directly inputting the image into MLLM causes the hidden caterpillar being incorrectly predicted as a leaf. Splitting the image results in interested objects being incomplete or absent, prompting MLLM to induce hallucinations and utilize prior knowledge to predict potential task-related objects within the image. Our visual contrastive reasoning then eliminates the hallucinations and validates the gathered predictions, aiding in the accurate identification of the caterpillar.

A task-generic prompt is both coarse and potentially ambiguous, can result in poor segmentation when directly applied. To address this problem, existing methods [21, 38] utilize the prior knowledge embedded in Multimodal Large Language Models (MLLMs) to infer more detailed, instance-specific prompts, such as bounding boxes or keywords, to guide the segmentation. However, these MLLMs often generate hallucinations due to object co-occurrence priors [68, 65], mistakenly predicting non-existent elements based on the environment as instance-specific prompts (Fig.1(a)). This can mislead segmentation and degrade model performance. While it is common to consider MLLM's hallucinations as detrimental and should be eradicated [52], this phenomenon actually demonstrates a MLLM's significant capacity for contextual inference based on prior training. We want to explore MLLM hallucinations as a valuable untapped knowledge resource for scene understanding, critical in complex segmentation scenarios. In practice, when task-related objects are not prominently visible, hallucinations can fill in missing information with plausible predictions based on learned patterns of association. Moreover, they can also extend beyond these familiar patterns, exploring and identifying new relationships within the data that were not explicitly taught during training. This dual ability to replicate and innovate makes hallucinations a valuable asset for enhancing model performance in complex or new situations. This predictive reasoning capacity not only fills perceptual gaps but also enriches the model's understanding, as hallucinations utilize prior knowledge to replicate and discover new patterns, enhancing insight into the target domain (see Fig.1(b)). Despite the potential benefits, using hallucinations to extract useful information from images to aid task remains unexplored.

In this work, instead of direct eliminating hallucinations, we utilize them as prior knowledge to mine extended task-related information from a given test image, performing scene understanding on the image before segmentation, then systematically reduce irrelevant hallucinations iteratively by visual masking verification, optimizing jointly instance-specific prompts and masks. To this end, we introduce an iterative, training-free Prompt-Mask Cycle Generation method (ProMaC) that refines segmentation through cyclic interactions between a prompt and mask generator (see Fig. 2). The prompt generator uses a multi-scale chain-of-thought prompting mechanism, which utilizes hallucinations to hypothesize and visual masking to verify, thereby creating more accurate instance-specific prompts. We trigger the hallucinatory tendencies of MLLMs, the process starts by dividing the image into patches at different scales and positions. Such partial visibilities of objects facilitate MLLMs to hypothesize potential object semantic labels and visual locations based on its prior knowledge. For validating the correctness of these hypotheses, we formulate a visual contrastive reasoning mechanism to generate contrastive images that contain only the background without any potential task-related objects. This helps identify all possible co-occurrence hallucinations caused by the background. By comparing these contrastive images with the original images, the MLLM effectively distinguishes between accurate hypotheses and those influenced by misleading prior knowledge, leading to more reliable prompts. Given the current promptable segmentation models' strength at mask prediction but struggle with label prediction, the mask generator uses mask semantic alignment to ensure that the produced masks align with the task semantics. These aligned masks not only serve as outputs but also guide the prompt generator in subsequent cycles, enhancing both prompt and mask quality continuously. **Our contributions are three-folds:**

1). We introduce a training-free Prompt-Mask Cycle Generation (ProMaC) to perform two tasks: Explore MLLM hallucinations as prior knowledge to enhance contextual scene understanding on each test image; systematically reduce irrelevant hallucinations to verify iteratively and optimize jointly both generated prompts and visual masking in object segmentation.

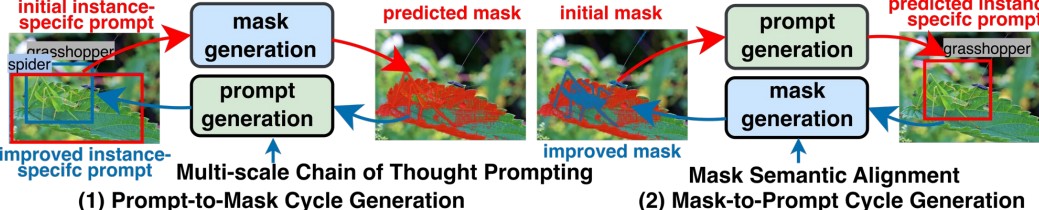

Figure 2: An overview of ProMac: Masks created iteratively by the mask generator guide the prompt generator to jointly improve instance-specific prompts and visual masking in segmentation.

2). We formulate an iterative optimization method including a prompt generator and a mask generator. To improve prompt relevance, the prompt generator utilizes a multi-scale chain of thought approach. It first leverages hallucinations to expand task-related plausible prompts, then applies visual contrastive reasoning to validate and reduce irrelevant prompts. ProMaC's mask generator overcomes SAM's shortcomings in label prediction by creating masks that align semantically with generated prompts.

3). Comprehensive comparative evaluations on 5 different segmentation tasks with 12 diverse datasets against 22 existing models demonstrate the effectiveness of ProMaC.

## 2   Related Works

**Promptable Segmentation** refers to object segmentation with active interactions from user inputs. Interaction methods vary from points, boxes, to scribbles. SAM [29], AV-SAM[41], GroundingSAM [38] and SEEM [71] accept video, audio, and multimodal inputs. However, they often rely on manual prompts, which can be unclear and subjective. Even with these prompts, they typically excel only in specific tasks. To address this issue, GenSAM [21] introduces a manual-free promptable segmentation setting, where only one task-generic prompt is provided. This prompt can be applied to all images within the task for instance-specific segmentation without any additional manual prompting. GenSAM primarily utilizes MLLM to infer the names of task-related objects in the images and uses them as instance-specific prompts for SAM to guide segmentation. However, GenSAM lacks spatial information about objects and may lead to inaccurate prompt predictions in complex scenes.

**Hallucinations in MLLMs** refers to models generate content that does not exist in the input data [65]. This issue often arises from the models leveraging extensive prior training rather than just the immediate input, leading to false predictions on fine-grained details. There are some efforts to mitigate this problem, including refining training processes [55, 44] and improving model architectures [4]. Other efforts focus on aligning model outputs more closely with actual data, employing feedback mechanisms for real-time adjustments [54]. While current works focus on eliminating hallucinations to enhance performance [31, 64], our work explores how to utilize hallucinations to expand and reason plausible context and validate them iteratively to remove irrelevant generalizations.

**Visual Marking for MLLMs** has been explored in recent research to prompt MLLMs through manipulation of visual inputs: (i) adding learnable soft tokens to visual inputs for efficient parameter tuning [1, 28], (ii) using image sequences as demonstrations of a new task [2, 9], and (iii) overlaying visual markers like masks, boxes, and circles onto visual inputs to ground regions [61, 54]. Our work falls into the third category, employing visual guidance for reasoning. Yang et al. [59] propose set-of-mark (SoM) prompts, where images are segmented and numbered regions to improve GPT-4V [43] visual grounding. However, as detailed in Tab.5, we confirm previous findings [5] that this visual marker approach struggles with open-source MLLMs like LLaVA. Instead of proprietary models [54] or fine-tuning [5, 8, 23], our training-free ProMaC uses inpainting task-related regions and contrasting model output distributions to prompt MLLMs.

## 3   Methodology

We introduce ProMaC, a cycle-generation method for segmenting unknown multiple classes of objects training-free with only a single task-generic prompt. Specifically, given an image $X \in \mathbb{R}^{H \times W \times 3}$ from a test set, ProMaC employs a task-generic prompt $P_g$ across datasets in the same task to produce

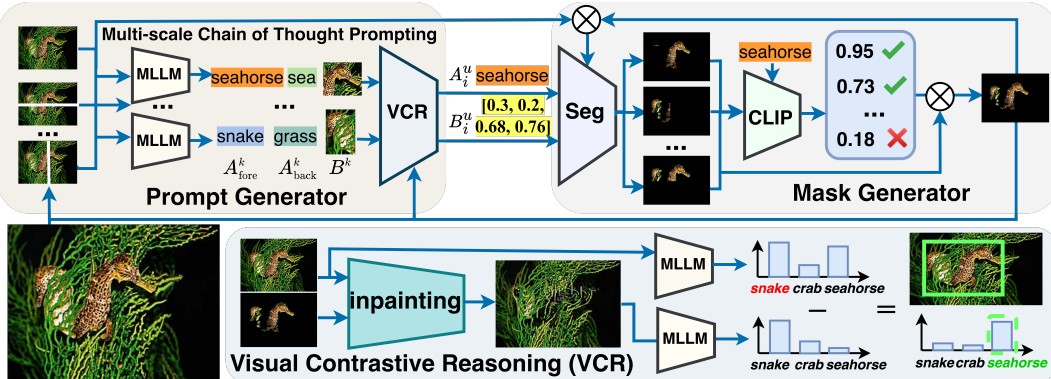

Figure 3: ProMaC consists of a prompt generator and a mask generator for cyclical optimization. The prompt generator employs multi-scale chain-of-thought prompting. It initially use hallucinations for exploring task-related information within image patches. It identifies task-relevant objects and their backgrounds ($A_{\text{fore}}^k$, $A_{\text{back}}^k$) along with their locations ($B^k$). Subsequently, it uses visual contrastive reasoning to refine and finalize instance-specific prompts ($A_i^u$, $B_i^u$) by eliminating hallucinations. The mask generator then processes these prompts into the segmentation model ("Seg"), producing a mask aligned with task semantics. This mask further guides the visual contrastive reasoning process, which leverages an inpainting model to eliminate masked regions, creating contrastive images. These images enable the prompt generator to further refine its prompts, enhancing segmentation accuracy.

a final segmentation mask $M \in \mathbb{R}^{H \times W}$, thereby removing the need for individual supervision for each image. The prompt generator leverages prior knowledge gained to reason and deduce instance-specific prompts, which then guide a mask generator to create masks aligned with task semantics. These masks act both as the current segmentation outcome and as visual markers for generating subsequent prompts. Training-free ProMaC relies solely on test-time adaptation.

## 3.1 Prompt Generator

Prompt generator employs MLLMs to generate instance-specific prompts based on image content and prior knowledge. It transforms the general prompt $P_g$, into an instance-specific prompt for each individual instance, providing more detailed descriptions of task-relevant objects. MLLM with parameters $\theta$ receives an image $X$ and query $P$ as inputs. $X$ provides contextual visual information to assist the model in generating a relevant response $y$ to the query $P$. The response $y$ is sampled auto-regressively from the probability distribution conditioned on $P$ and $X$ as follows:

$$y_t \sim p_\theta(y_t \mid X, P, y_{<t}) \propto \exp(\text{logit}_\theta(y_t \mid X, P, y_{<t})) \tag{1}$$

where $y_t$ denotes the token at time step $t$, and $y < t$ represents the sequence of generated tokens up to the time step $(t-1)$. In practice, predicted task-relevant objects can often blend into the background due to texture, color, size, or position, leading to inaccuracies in instance-specific prompts. To address this problem, we explore MLLM hallucinations as contextual prior knowledge from pretraining, rather than eliminate them. These hallucinations are particularly useful when direct visual cues are absent or ambiguous, helping the model fill in information gaps and hypothesize potential task-related elements within the image that are not prominent. By revealing these often-overlooked subtle associations, hallucinations provide a more comprehensive scene understanding of the image content. This deeper contextual understanding provide a reasoning context for generating more accurate and relevant instance-specific prompts candidates.Thus, using hallucinations to uncover task-related knowledge helps overcome challenges from visual ambiguities and object camouflage in complex scenes. To this end, we propose a multi-scale chain-of-thought prompting strategy that stimulates hallucinations to leverage prior knowledge, fully extracts task-relevant information, and then uses this information to enhance the precision of the generated instance-specific prompts.

### 3.1.1 Multi-scale Chain of Thought Prompting

Multi-scale Chain of Thought Prompting consists of two processes: Gathering candidate knowledge and generating accurate instance-specific prompts. To efficiently collect task-relevant information from an image, as shown in Fig. 3, we divide the input image into patches at various scales by cutting

horizontally, vertically, or by leaving it whole. These patches are then processed by the MLLM to gather preliminary instance-specific prompts. The differing levels of task-relevant object visibility in each patch prompt the MLLM to induce hallucinations. These hallucinations utilize prior knowledge to explore connections between the image data and the associated task, aiding in the detection of potential bounding boxes and object names. The process is computed by:

$$B^k = \text{MLLM}\left(X^k, C^k, P_B\right), \qquad A^k_{\text{fore}}, A^k_{\text{back}} = \text{MLLM}\left(X^k, C^k, P_A\right), \qquad (2)$$

where $C^k$ is the caption generated by MLLM for the $k-$th image patch $X^k$. $P_g$ is task-generic prompt. For bounding box prediction, the prompt $P_B$, which instructs "*This image is from the $P_g$ detection task, output the bounding box of the $P_g$.*". This guides the MLLM to predict the bounding box $B^k$ of the task-related objects within the patch. For predicting name, the prompt $P_A$, stating "*Output the name of the $P_g$ and its environment in one word.*" is used, guiding the MLLM to predict the names of the task-related objects $A^k_{\text{fore}}$ and their backgrounds $A^k_{\text{back}}$ from each patch. The preliminary data, including object names $A^k_{\text{fore}}$ and bounding boxes $B^k$, gathered from various patches, are compiled into candidate lists $\mathcal{A}_i$ and $\mathcal{B}_i$. Here, $i$ denotes the iteration in the iterative learning cycle. In this process, the hallucinations employed are essentially based on object co-occurrence priors, where objects commonly associated with background elements during pre-training are predicted to be task-relevant, even if they are not present in the current image. This prior knowledge is useful during the knowledge collection stage as it uncovers implicit relationships and details in the image. However, it can also reduce accuracy of the later fine-grained instance-specific prompts generation. Therefore, it is crucial to control these hallucinations in the latter stage to prevent incorrect predictions.

**Visual Contrastive Reasoning.** To mitigate hallucinations caused by object co-occurrence priors, recent research highlights particularly relevant regions of an image to direct MLLMs focus toward task-related elements, thereby minimizing background interference and enhancing model accuracy [59, 54]. To achieve this, visual markers are employed to steer MLLM attention on task-relevant visual regions, thereby reducing hallucinations. While closed-source MLLMs like GPT-4V [43] can interpret these markers effectively, they are costly and large. In contrast, models like LLaVA [37] are open-source, but cannot process visual markers such as points or bounding boxes, and employing these markers might disrupt the original pixel data, degrading performance on LLaVA (see Tab. 5). Moreover, accurate pixel-level visual markers are unavailable in our setting. To solve this problem, we aim to enable LLaVA to focus on task-related regions without altering the original pixel data, thereby effectively minimizing hallucinations and enhancing the precision of instance-specific prompts.

Despite the absence of instance-level annotations, promptable segmentation models produce masks with detailed textures, which provide rich positional and textural information about interested regions. We use these masks as visual markers to guide a MLLM to focus on task-related areas during the generation of instance-specific prompts. Inspired by classifier-free guidance [20, 47], we introduce visual contrastive reasoning (VCR), a training-free visual marking method to help MLLM focus on specific regions, reducing hallucinations. The relevance of a region is assessed by observing MLLM output changes when key areas are excluded. It guides the MLLM to focus on areas with notable changes (bottom of Fig.3). Based on Eq. (1), we derive a probability distribution by comparing original image $X$ with a modified image, $X' = \text{process}(X, \text{IM})$, where region IM is excluded.

$$y_t \propto p_\theta(y_t \mid X, P, y_{<t}) \left( \frac{p_\theta(y_t \mid X, P, y_{<t})}{p_\theta(y_t \mid \text{process}(X, \text{IM}), P, y_{<t})} \right)^\alpha$$
$$\sim \text{softmax}[(1 + \alpha) \cdot \text{logit}_\theta(y_t \mid X, P, y_{<t}) - \alpha \cdot \text{logit}_\theta(y_t \mid \text{process}(X, \text{IM}), P, y_{<t})], \quad (3)$$

where $\alpha$ is the level of focus on region IM. A higher $\alpha$ increases emphasis on that region. Following [54], we set $\alpha = 1$ in all tasks. It preserves the integrity of the original image pixels $X$, while constructing contrastive samples $X'$ that encourage the model to focus on task-related regions. Ideally, $X'$ should exclude task-related objects while maintaining a uniform appearance and overall context with the original image. But directly marking $X'$ disrupts its pixels, making contrastive sample generation challenging.

**Contrastive Sample Generation.** To address it, we employ inpainting, where the mask $M_{i-1}$ obtained from the previous iteration segmentation is treated as the inpainting mask $\text{IM}_i$ to guide the creation of $X'$. We use a negative prompt $P_n$: "$A^{fore}_i$, *is a $P_g$*", to ensure that the inpainted $X'$ does not contain potentially task-related objects $A^{\text{fore}}_i$. Additionally, we use a positive prompt $P_p$: "$A^{back}_i$, *high quality, detailed, blended to the original image.*", to ensure consistency between the generated

portion and the surrounding background $A_i^{\text{back}}$. The corresponding inpainting is defined as:

$$X' = F_{in}(X, \text{IM}_i, P_p, P_n), \tag{4}$$

where $F_{in}$ represents the inpainting module, and we choose Stable Diffusion to perform this operation. This method ensures the generated $X'$ excludes task-related objects without disrupting the pixel continuity. In the first iteration, since $\text{IM}_i$ does not yet exist, we use bounding box predictions from various patches $\mathcal{B}_i$ as an alternative. As $X'$ contains only the background, comparing it with $X$ eliminates co-occurrence hallucination caused by the background and highlights differences in task-related regions, subtly guiding the model to focus on these areas. Finally, we use visual contrastive reasoning to identify accurate instance-specific prompt to guide segmentation as follows,

$$B_i^u = \text{VCR}\left(X, X', C, P_B\right), \qquad A_i^u = \text{VCR}\left(X, X', C, P_A\right), \tag{5}$$

where VCR represents our visual contrastive reasoning, and $C$ is the caption of the image. The collected knowledge, $\mathcal{A}i$ and $\mathcal{B}i$, is integrated into the prompt $P_A$ and $P_B$. This process aids in identify the ultimate instance-specific names $A_i^u$ and bounding boxes $B_i^u$ of the objects.

## 3.2 Mask Generator

Until now, we described how to use SAM-generated masks as a visual marker to guide the model to focus on task-relevant areas for generating accurate instance-specific prompts. But this method relies on an assumption that the mask accurately delineates task-related regions. However, SAM is trained on large-scale prompt-mask pairs without category labels, it excels at identifying masks based on image textures but lacks label prediction capabilities. Consequently, the SAM-generated mask may not always align with task semantics, yet such alignment is crucial for our method.

### 3.2.1 Mask Semantic Alignment

We need to utilize texture generalization capabilities of SAM to describe possible task-related objects within the prompt-targeted areas, while also ensuring that the generated masks align with the task semantics. To achieve this, we divide the input image into patches of varying scales using horizontal, vertical, and uncut divisions as outlined in the last section. these processed patches are then reintegrated onto the original image with surrounding areas blacked out, and fed into SAM to ensure it focuses exclusively on the patch. Finally, masks generated from different patches are aggregated based on their relevance to task semantics, providing an accurate representation of task-related objects. The masks for each patch is generated as follows,

$$m_i^k = \text{SAM}(\text{Spatial CLIP}(A_i^u, X_i), B_i^u, X_i^k), \tag{6}$$

where mask $m_i^k$ is obtained by inputting the corresponding image patch $X_i^k$ and associated prompts into SAM during the $i-$th iteration. Following [21], Spatial CLIP maps the text prompt $A_i^u$ to regions in the image $X_i$ that correspond to the content of the prompt. The processed images, along with the generated instance-specific text prompts $A_i^u$, are then input into CLIP to assess semantic similarity.

$$s(m_i^k) = \text{CLIP}(m_i^k \odot X_i, A_i^u), \tag{7}$$

the operation $\odot$ results in retaining only those parts of $X_i$ that are covered by the predicted mask. $s(m_i^k)$ represents the similarity between masked image and $A_i^u$, calculated using CLIP. The similarity scores obtained from different patches are denoted as $S_i = [s(m_i^1), s(m_i^2), \ldots, s(m_i^k)]$. After normalizing the elements within $S_i$, the closer the normalized $s(m_i^k)$ is to 1, the more semantically aligned $m_i^k$ is with the instance-specific text prompt $A_i^u$. Finally, we compute the weighted sum of the normalized $s(m_i^k)$ and $m_i^k$ as follows.

$$M_i = \sum_{k=1}^{K} (s(m_i^k) * m_i^k), \tag{8}$$

$M_i$ is the output mask of the $i-$th iteration of $X$. The generated $M_i$ leverages SAM's mask prediction capabilities to create highly detailed masks. Simultaneously, through the mask semantic alignment process, it ensures that the output mask aligns with the task's semantics, thereby overcoming the limitation of SAM's mask prediction lacking semantic understanding. The mask is applied to the original image as a weight, to generate the next iteration image $X_i$ for segmentation. This excludes irrelevant regions to reduce interference during segmentation.

$$X_{i+1} = w \cdot (X_i \odot M_i) + (1 - w) \cdot X_i, \tag{9}$$

where $w$ is a hyperparameter, which we have assigned a value of 0.3.

Table 1: Results on Camouflaged Object Detection (COD) under different settings. Best are in **bold**.

| Methods | Venue | Camouflaged Object Detection | | | | | | | | | | | |
|---|---|---|---|---|---|---|---|---|---|---|---|---|---|
| | | CHAMELEON [50] | | | | CAMO [30] | | | | COD10K [14] | | | |
| | | $M\downarrow$ | $F_\beta\uparrow$ | $E_\phi\uparrow$ | $S_\alpha\uparrow$ | $M\downarrow$ | $F_\beta\uparrow$ | $E_\phi\uparrow$ | $S_\alpha\uparrow$ | $M\downarrow$ | $F_\beta\uparrow$ | $E_\phi\uparrow$ | $S_\alpha\uparrow$ |
| Scribble Supervision Setting | | | | | | | | | | | | | |
| WSSA[62] | CVPR20 | 0.067 | 0.692 | 0.860 | 0.782 | 0.118 | 0.615 | 0.786 | 0.696 | 0.071 | 0.536 | 0.770 | 0.684 |
| SCWS[60] | AAAI21 | 0.053 | 0.758 | 0.881 | 0.792 | 0.102 | 0.658 | 0.795 | 0.713 | 0.055 | 0.602 | 0.805 | 0.710 |
| TEL[62] | CVPR22 | 0.073 | 0.708 | 0.827 | 0.785 | 0.104 | 0.681 | 0.797 | 0.717 | 0.057 | 0.633 | 0.826 | 0.724 |
| SCOD[17] | AAAI23 | **0.046** | **0.791** | **0.897** | 0.818 | **0.092** | 0.709 | 0.815 | 0.735 | 0.049 | 0.637 | 0.832 | 0.733 |
| SAM-S[29] | ICCV23 | 0.076 | 0.729 | 0.820 | 0.650 | 0.105 | 0.682 | 0.774 | 0.731 | 0.046 | 0.695 | 0.828 | 0.772 |
| WS-SAM[16] | NeurIPS23 | **0.046** | 0.777 | **0.897** | 0.824 | **0.092** | 0.742 | 0.818 | 0.759 | 0.038 | 0.719 | 0.878 | 0.803 |
| Point Supervision Setting | | | | | | | | | | | | | |
| WSSA[62] | CVPR20 | 0.105 | 0.660 | 0.712 | 0.711 | 0.148 | 0.607 | 0.652 | 0.649 | 0.087 | 0.509 | 0.733 | 0.642 |
| SCWS[60] | AAAI21 | 0.097 | 0.684 | 0.739 | 0.714 | 0.142 | 0.624 | 0.672 | 0.687 | 0.082 | 0.593 | 0.777 | 0.738 |
| TEL[62] | CVPR22 | 0.094 | 0.712 | 0.751 | 0.746 | 0.133 | 0.662 | 0.674 | 0.645 | 0.063 | 0.623 | 0.803 | 0.727 |
| SCOD[17] | AAAI23 | 0.092 | 0.688 | 0.746 | 0.725 | 0.137 | 0.629 | 0.688 | 0.663 | 0.060 | 0.607 | 0.802 | 0.711 |
| SAM[29] | ICCV23 | 0.207 | 0.595 | 0.647 | 0.635 | 0.160 | 0.597 | 0.639 | 0.643 | 0.093 | 0.673 | 0.737 | 0.730 |
| SAM-P[29] | ICCV23 | 0.101 | 0.696 | 0.745 | 0.697 | 0.123 | 0.649 | 0.693 | 0.677 | 0.069 | 0.694 | 0.796 | 0.765 |
| WS-SAM[16] | NeurIPS23 | **0.056** | **0.767** | **0.868** | **0.805** | **0.102** | **0.703** | **0.757** | **0.718** | **0.039** | **0.698** | **0.856** | **0.790** |
| Task-Generic Prompt Setting | | | | | | | | | | | | | |
| CLIP_Surgey+SAM | Arxiv23 | 0.147 | 0.606 | 0.741 | 0.689 | 0.189 | 0.520 | 0.692 | 0.612 | 0.173 | 0.488 | 0.698 | 0.629 |
| GPT4V+SAM [43, 29] | Arxiv23 | 0.180 | 0.557 | 0.710 | 0.637 | 0.206 | 0.466 | 0.666 | 0.573 | 0.187 | 0.448 | 0.672 | 0.601 |
| LLaVA1.5+SAM [37, 29] | NeurIPS23 | 0.168 | 0.561 | 0.718 | 0.666 | 0.314 | 0.401 | 0.585 | 0.501 | 0.170 | 0.530 | 0.728 | 0.662 |
| X-Decoder [69] | CVPR23 | 0.124 | 0.654 | 0.748 | 0.716 | 0.104 | 0.628 | 0.745 | 0.709 | 0.171 | 0.556 | 0.705 | 0.652 |
| SEEM [71] | NeurIPS23 | 0.094 | 0.011 | 0.307 | 0.454 | 0.192 | 0.023 | 0.315 | 0.404 | 0.143 | 0.001 | 0.280 | 0.425 |
| GroundingSAM [29, 38] | ICCV23 | 0.122 | 0.662 | 0.776 | 0.744 | 0.157 | 0.656 | 0.753 | 0.707 | 0.085 | 0.670 | 0.813 | 0.764 |
| GenSAM [21] | AAAI24 | 0.073 | 0.696 | 0.806 | 0.774 | 0.106 | 0.669 | 0.798 | 0.729 | 0.058 | 0.695 | 0.843 | 0.783 |
| ProMaC | Ours | **0.044** | **0.790** | **0.899** | **0.833** | **0.090** | **0.725** | **0.846** | **0.767** | **0.042** | **0.716** | **0.876** | **0.805** |

## 3.3 Mask Prompt Cycle Generation

The mask generated from the last iteration will guide the prompt generator in the next iteration to focus on potential task-related regions, eliminating the erroneous effects of irrelevant hallucinations and generating more accurate instance-specific prompts. These prompts, in turn, help the mask generator produce better masks. Through iterative prompt generation and mask generation jointly, we yield both better instance-specific prompts and visual masks. Finally, the masks from different iterations are averaged, and the mask closest to the mean is considered the final output.

$$\mathrm{i}^* = \arg\min_i \left( \left| M_i - \frac{\sum_i (M_1, \ldots, M_\mathbf{I})}{i_{\text{result}}} \right| \right). \tag{10}$$

Here, $\mathbf{I}$ is the number of adaptation epochs and $M_{\mathrm{i}^*}$ is the corresponding final mask for image $X$.

Table 2: Results for Medical Image Segmentation (MIS) under task-generic prompt setting.

| Methods | Venue | Polyp Image Segmentation | | | | | | | | Skin Lesion Segmentation | | | |
|---|---|---|---|---|---|---|---|---|---|---|---|---|---|
| | | CVC-ColonDB [51] | | | | Kvasir [25] | | | | ISIC [10] | | | |
| | | $M\downarrow$ | $F_\beta\uparrow$ | $E_\phi\uparrow$ | $S_\alpha\uparrow$ | $M\downarrow$ | $F_\beta\uparrow$ | $E_\phi\uparrow$ | $S_\alpha\uparrow$ | $M\downarrow$ | $F_\beta\uparrow$ | $E_\phi\uparrow$ | $S_\alpha\uparrow$ |
| GPT4V+SAM [43, 29] | Arxiv23 | 0.578 | 0.051 | 0.246 | 0.242 | 0.614 | 0.128 | 0.236 | 0.253 | 0.514 | 0.387 | 0.366 | 0.334 |
| LLaVA1.5+SAM [37, 29] | NeruIPS23 | 0.491 | 0.194 | 0.355 | 0.357 | 0.479 | 0.293 | 0.400 | 0.403 | 0.369 | 0.473 | 0.497 | 0.477 |
| X-Decoder [69] | CVPR23 | 0.462 | 0.095 | 0.327 | 0.331 | 0.449 | 0.202 | 0.371 | 0.384 | 0.338 | 0.315 | 0.127 | 0.407 |
| SEEM [71] | NeruIPS23 | 0.570 | 0.085 | 0.280 | 0.284 | 0.520 | 0.215 | 0.339 | 0.367 | 0.362 | 0.250 | 0.002 | 0.280 |
| GroundingSAM [29, 38] | ICCV23 | 0.711 | 0.071 | 0.195 | 0.206 | 0.387 | 0.353 | 0.521 | 0.468 | 0.301 | 0.348 | 0.247 | 0.533 |
| GenSAM [21] | AAAI24 | 0.244 | 0.059 | 0.494 | 0.379 | 0.172 | 0.210 | 0.619 | 0.487 | 0.171 | 0.699 | 0.744 | 0.678 |
| ProMaC | Ours | **0.176** | **0.243** | **0.583** | **0.530** | **0.166** | **0.394** | **0.726** | **0.573** | **0.168** | **0.717** | **0.755** | **0.689** |

Table 3: Result on Transparent Object Segmentation and Open-Vocabulary Segmentation Tasks.

(a) Transparent Object Segmentation.

| Methods | GSD [34] | | | | Trans10K-hard [56] | | | |
|---|---|---|---|---|---|---|---|---|
| | $M\downarrow$ | $F_\beta\uparrow$ | $E_\phi\uparrow$ | $S_\alpha\uparrow$ | $M\downarrow$ | $F_\beta\uparrow$ | $E_\phi\uparrow$ | $S_\alpha\uparrow$ |
| GPT4V+SAM [43, 29] | 0.312 | 0.104 | 0.392 | 0.363 | 0.288 | 0.199 | 0.607 | 0.512 |
| LLaVA1.5+SAM [37, 29] | 0.197 | 0.202 | 0.545 | 0.433 | 0.272 | 0.167 | 0.621 | 0.555 |
| X-Decoder [69] | 0.191 | 0.240 | 0.643 | 0.480 | 0.568 | **0.611** | 0.218 | 0.280 |
| SEEM [71] | 0.184 | 0.224 | 0.573 | 0.479 | 0.557 | 0.501 | 0.013 | 0.256 |
| GroundingSAM [29, 38] | 0.168 | 0.230 | 0.572 | 0.483 | 0.436 | 0.415 | 0.047 | 0.424 |
| GenSAM [21] | 0.155 | 0.394 | 0.700 | 0.559 | 0.263 | 0.489 | 0.612 | 0.536 |
| ProMaC | **0.147** | **0.409** | **0.723** | **0.569** | **0.251** | 0.509 | **0.654** | **0.557** |

(b) Open-vocabulary Segmentation.

| Methods | Venue | Seg. Anno. | Image-Text pairs | VOC mIoU↑ | Context mIoU↑ | Object mIoU↑ |
|---|---|---|---|---|---|---|
| MaskCLIP[67] | ECCV22 | - | - | 38.8 | 23.6 | 20.6 |
| TCL [6] | CVPR23 | - | CC3M [48], CC12M [7] | 51.2 | 24.3 | **30.4** |
| GroupViT [57] | CVPR22 | - | CC12M [7], YFCC14M [53] | 52.3 | 22.4 | - |
| ViewCo [46] | ICLR23 | - | CC12M [7], YFCC14M [53] | 52.4 | 23.0 | 23.5 |
| SegCLIP [39] | ICML23 | COCO [35] | CC [48] | 52.6 | 24.7 | 26.5 |
| OVSegmentor [58] | CVPR23 | - | CC12M [7] | 53.8 | 20.4 | 25.1 |
| ProMaC | Ours | - | - | **59.3** | **30.7** | 25.2 |

## 4 Experimental Setup

**Baselines.** To evaluate our approach across various scenarios, we first assess the performance of ProMaC on challenging segmentation tasks, including Camouflaged Object Detection (COD), Medical

Table 4: Ablation Study on COD and MIS Tasks

| Method's Variants | | | | | CHAMELEON [50] | | | | CVC-ColobNB [51] | | | |
|---|---|---|---|---|---|---|---|---|---|---|---|---|
| MCoT | IVP | ITP | VCR | MSA | $M\downarrow$ | $F_\beta\uparrow$ | $E_\phi\uparrow$ | $S_\alpha\uparrow$ | $M\downarrow$ | $F_\beta\uparrow$ | $E_\phi\uparrow$ | $S_\alpha\uparrow$ |
| | ✓ | ✓ | ✓ | ✓ | 0.052 | 0.764 | 0.885 | 0.816 | 0.187 | 0.214 | 0.570 | 0.513 |
| ✓ | | | ✓ | ✓ | 0.080 | 0.720 | 0.833 | 0.757 | 0.260 | 0.123 | 0.466 | 0.425 |
| ✓ | ✓ | | ✓ | ✓ | 0.089 | 0.685 | 0.823 | 0.756 | 0.177 | 0.233 | 0.556 | 0.524 |
| ✓ | ✓ | ✓ | | ✓ | 0.061 | 0.769 | 0.893 | 0.815 | 0.311 | 0.152 | 0.460 | 0.424 |
| ✓ | ✓ | ✓ | ✓ | | 0.054 | 0.740 | 0.884 | 0.798 | **0.156** | 0.220 | 0.565 | 0.517 |
| ✓ | ✓ | ✓ | ✓ | ✓ | **0.044** | **0.790** | **0.899** | **0.833** | 0.176 | **0.243** | **0.583** | **0.530** |

Table 5: VCR Result on SR task

| Model | Indiv. | Pairs | Set of 4 |
|---|---|---|---|
| CLIP ViT-L-14 [45] | 26.1 | 1.5 | 0.0 |
| CLIP RN50x64 [45] | 26.2 | 2.0 | 0.0 |
| FLAVA [49] | 30.4 | 10.9 | 0.0 |
| ViP-LLAVA-13B [5] | 70.9 | 57.5 | 21.8 |
| LLAVA-1.5-13B [36] | 73.1 | 60.6 | 28.9 |
| + VCR (Ours) | **75.4** | **63.6** | **36.7** |

Table 6: Parameter ablation study on COD10K [14].

(a) Number of iteration $\mathbf{I}$.

| $\mathbf{I}$ | cos↑ | IoU↑ | $M\downarrow$ | $F_\beta\uparrow$ | $E_\phi\uparrow$ | $S_\alpha\uparrow$ |
|---|---|---|---|---|---|---|
| 1 | 0.864 | 0.563 | 0.080 | 0.626 | 0.818 | 0.765 |
| 2 | 0.876 | 0.589 | 0.050 | 0.683 | 0.859 | 0.796 |
| 3 | 0.879 | 0.593 | 0.045 | 0.702 | 0.869 | 0.802 |
| 4 | **0.882** | 0.601 | 0.042 | 0.714 | 0.875 | **0.804** |
| 5 | 0.881 | **0.602** | **0.041** | 0.718 | 0.875 | **0.804** |
| 6 | **0.882** | 0.599 | **0.041** | 0.721 | 0.876 | 0.803 |

(b) Image preprocess strategy.

| Scale | $M\downarrow$ | $F_\beta\uparrow$ | $E_\phi\uparrow$ | $S_\alpha\uparrow$ |
|---|---|---|---|---|
| Original | 0.075 | 0.535 | 0.750 | 0.662 |
| Havel | 0.069 | 0.579 | 0.775 | 0.689 |
| Quarters | 0.087 | 0.423 | 0.673 | 0.586 |
| Original+Havel | **0.042** | **0.714** | **0.875** | **0.804** |
| Original +Havel+Quarters | 0.049 | 0.702 | 0.867 | 0.796 |

(c) Visual marker strategy.

| strategy | $M\downarrow$ | $F_\beta\uparrow$ | $E_\phi\uparrow$ | $S_\alpha\uparrow$ |
|---|---|---|---|---|
| None | 0.058 | 0.690 | 0.855 | 0.789 |
| Bbox | 0.065 | 0.682 | 0.836 | 0.766 |
| VCD | 0.047 | 0.705 | 0.863 | 0.793 |
| Ours | **0.042** | **0.714** | **0.875** | **0.804** |

Image Segmentation (MIS), and Transparent Object Detection (TOD). These tasks are areas where SAM struggle [26]. In the COD task, we compare ProMaC with weakly supervised segmentation methods [29, 62, 60, 62, 17, 17, 22, 24, 18]. Two supervision levels are used for comparison: scribble supervision, where main structures for the foreground and background are sketched during training, and point supervision, where separate points are provided for both foreground and background. In the task-generic prompt setting, we introduce a challenging scenario by only providing a task description as a generic prompt for segmentation. ProMaC integrates LLaVA1.5 [37] with SAM [29]. We also experiment on the MIS and PIS tasks to demonstrate the effectiveness of our method using task-generic prompts compared to previous methods. We assess GPT4V+SAM and LLaVA1.5+SAM in this setting to demonstrate that current MLLM models cannot address it well. We also compare ProMaC against current SOTA promptable segmentation methods to showcase its effectiveness. Next, we evaluate ProMaC on Open-Vocabulary Segmentation (OVS), and compare with leading methods [33, 70, 71, 38, 63, 66, 32, 19]. Our Visual Contrastive Reasoning (VCR) strategy is applicable to other tasks, especially those requiring complex spatial understanding. We used the What's Up spatial reasoning dataset [27] to evaluate how well VCR guides models to focus on task-relevant regions in Spatial Reasoning (SR) task. Our results are the average of three trials.

**Metric.** For evaluating the first three tasks, metrics Mean Absolute Error (M), adaptive F-measure ($F_\beta$) [40], mean E-measure ($E_\phi$) [15], and structure measure ($S_\alpha$) [13] are used. Lower M or higher values for $F_\beta, E_\phi$, and $S_\alpha$ reflect better performance. Mean Intersection over Union (mIoU) and accuracy measure OVS and SR performance respectively, with higher values indicating better results.

**PyTorch Implementation Details.** For the MLLM models, we utilize LLaVA-1.5-13B for evaluation purposes. For CLIP, we adopt the CS-ViT-B/16 pretrained model. The inpainting model is stable-diffusion-2-inpainting. The task-generic prompts for the COD task is "camouflaged animal". The MIS task consists of two sub-tasks: polyp image segmentation and skin lesion segmentation, each with its own task-generic prompts, "polyp" and "skin lesion" respectively. For TOD task, prompt is fixed as "glass". All tasks are optimized using training-free test-time adaptation, with each task iterating for four epochs, except for the polyp image segmentation task, which undergoes six epochs. Since the second epoch, VCR also operates on different patches, ensuring that while the non-inpainted parts still gather information through hallucinations, the inpainted parts eliminate hallucinations to generate accurate candidate prompts. The promptable segmentation methods is the ViT-H/16 version of SAM. Our experiment is conducted on a single NVIDIA A100 GPU. More details are in appendix.

## 5 Results and Analysis

**Results on COD Task.** The COD tasks focus on finding animals that blend into their complex surroundings. We evaluated ProMaC on three representative datasets: CHAMELEON [50], CAMO [30], and COD10K [14]. As shown in Tab. 1, we compared ProMaC with others that utilize varying levels of supervision. Overall, methods with scribble supervision generally perform better than those with point supervision. Importantly, ProMaC only uses a single generic task prompt for the entire task and it stil outperforms all point-supervised methods. It also surpasses methods with scribble supervision on the CHAMELEON and CAMO datasets, and matches the top-performing scribble-supervised methods on COD10K. It demonstrates the superiority of ProMaC.

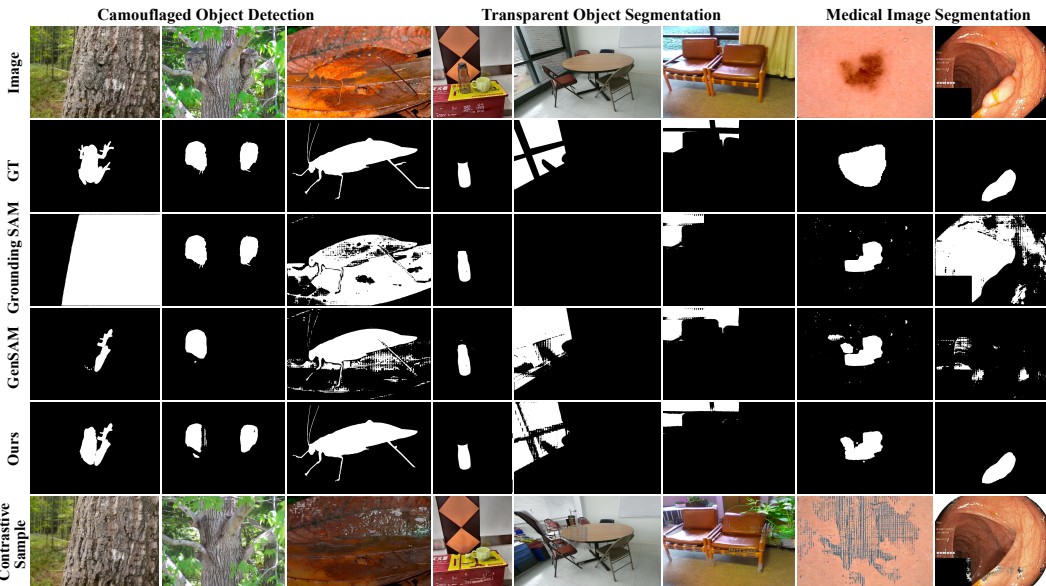

Figure 4: Visualization of various segmentation methods among various segmentation tasks.

**Results on MIS and TOD Task.** The MIS task identifies pathological tissues in medical images. We used three datasets: ColonDB [51] and Kvasir [25] for polyp image segmentation, and ISIC [10] for skin lesion segmentation. We compared our approach with others using task-generic prompt settings (see Tab. 2). While other models underperform in medical imaging due to limited generalization, ProMaC improves significantly over the baseline by iteratively mining task-related knowledge. For the TOD task, we evaluated ProMaC on the GSD [34] and Trans10K-hard [56] datasets (See Tab. 3(a)). Using the task-generic prompt setting, our method achieves the best results despite challenging scenarios. This demonstrates ProMaC's versatility and adaptability across complex visual tasks.

**Results on OVS and SR Task.** We evaluated ProMaC's effectiveness on the OVS task for multi-class segmentation based on a list of candidate classes. Specifically, we tested it on the validation splits of PASCAL VOC (21 classes) [12, 11], Pascal Context (59 classes) [42], and COCO-Object (80 classes) [3], using LLaVA to identify and confirm the presence of candidate classes. After obtaining masks, we resolved overlaps using the argmax operation based on SAM probabilities. Tab. 3(b) shows how ProMaC compares to other state-of-the-art OVS methods. Unlike some methods trained specifically on these datasets (risking knowledge leaking), ProMaC is not. Yet, ProMaC still outperforms all others on PASCAL VOC and Pascal Context and is competitive on COCO-Object. Additionally, as shown in Tab. 5, we integrated our VCR into LLaVA1.5 for enhanced spatial reasoning. This integration allows LLaVA to better focus on critical areas, thereby boosting performance.

**Module Analysis.** As shown in Tab. 4, we perform an ablation study on the COD and MIS tasks to assess the effects of different modules. "MCoT" is multi-scale chain of thought prompting. "ITP" and "IVP" refer to using only instance-specific text prompts or visual prompts. "VCR" is visual contrastive reasoning, and "MSA" is mask semantic alignment. The first row shows replacing MCoT with just one original image results in reduced performance, highlighting the importance of using hallucinations to extract task-relevant information. The second and third rows show that single modal prompts perform worse than multimodal prompts, highlighting the significance of multimodal prompting. Removing VCR causes a significant drop in performance, indicating that visual prompts are crucial for directing LLaVA's focus on relevant areas during inference. The comparison between the fifth and final rows emphasizes the importance of mask alignment with task semantics. The consistent positive results across tasks confirm the robustness and effectiveness of our approach.

**Parameter Analysis.** Tab. 6(a) examines how iterations influence performance. "cos" measures the cosine similarity between the predicted text prompt and the ground truth class through CLIP. "IoU" assesses the overlap between the predicted bounding box and the ground truth, comparing it against a rectangular outline of the mask. Mask predictions improve and stabilize after the fourth epoch. Tab. 6(b) investigates the effects of various image processing techniques. "Original" uses no modifications,

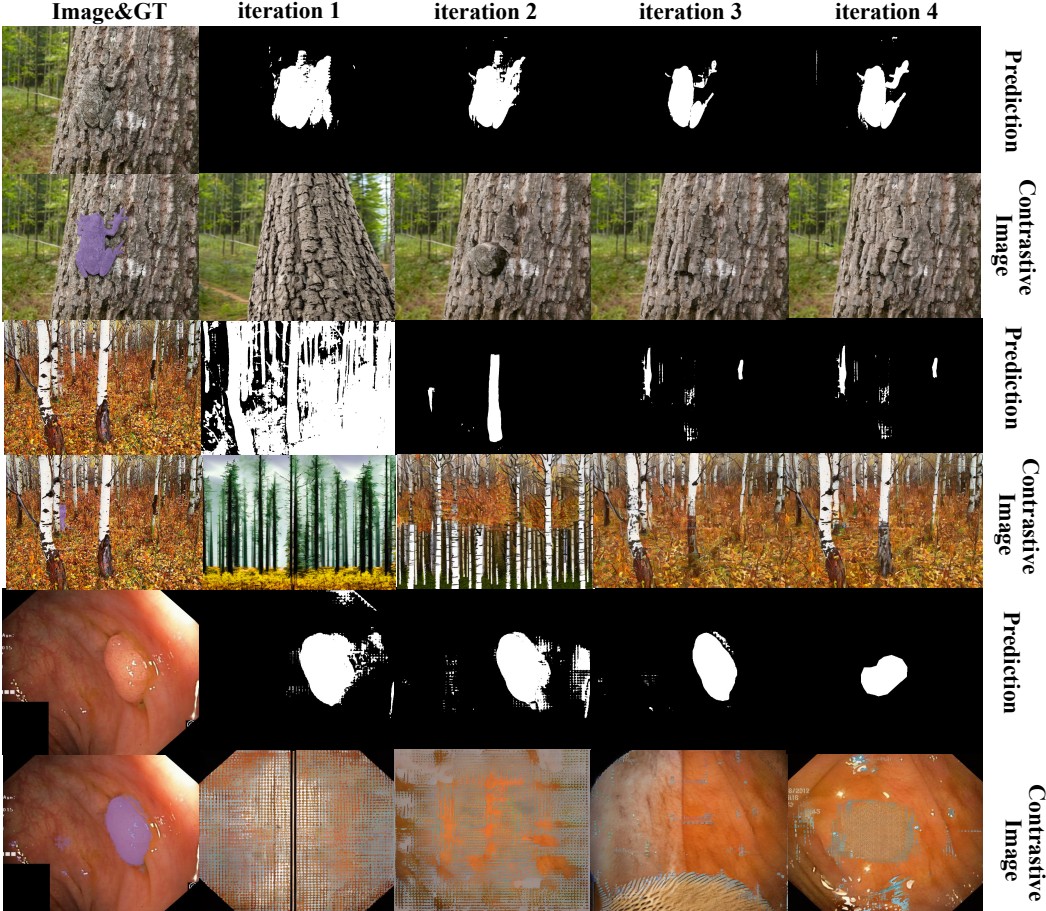

Figure 5: Visualization of the generated masks and contrastive samples over iterations.

"Halve" divides the image horizontally or vertically into halves, and "Quarters" divides it into four quarter-sized patches. Testing shows that combining "Original" and "Halve" yields the best results by balancing global and local information without excessive fragmentation.

**Visual Marker Strategy.** Tab. 6(c) assesses the impact of different visual marker strategies. "None" uses no visual prompts, while "Bbox" places bounding boxes directly on the image. "VCD" employs previous methods that introduce Gaussian noise into comparison images for contrastive reasoning. Results indicate that bounding boxes decrease performance, suggesting LLaVA struggles with this type of markers. Although VCD methods improve performance, they distort pixel data, making them less effective than our approach. Our VCR generates contrastive samples that focus on task-relevant areas without altering the image, reducing hallucinations and enhancing performance.

**Visualization.** Fig. 4 and Fig.5 visually compares our ProMaC with other methods across 3 tasks and also shows the contrastive images we generated. GenSAM handles clear objects well but struggles with complex background. Although GenSAM performs well in complex backgrounds, but struggles with challenging tasks. ProMaC delivers solid segmentation results across different tasks, and our contrastive images remove task-related regions while maintaining semantic and pixel consistency.

## 6   Conclusion

In this work, we introduce an iterative ProMaC that uses MLLM hallucinations to guide automatic prompt generation, significantly improving segmentation without training. This iterative approach aligns masks with task semantics, enhancing model performance. Testing on multiple benchmarks has demonstrated ProMaC's effectiveness in a wide range of complex segmentation tasks.

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

# A   Appendix

## A.1   More Explanation on Task-generic Promptable Segmentation

Task-generic promptable segmentation, first introduced by [21], aims to solve a key challenge in promptable segmentation: the requirement for manual prompts, such as bounding boxes, scribbles, or points, for each sample within the same task. This method allows a model, like the Segment Anything Model (SAM), to segment samples based on these prompts. Task-generic promptable segmentation seeks to enable models to automatically infer the task-related objects in different images based on a general task description. This approach eliminates the need to manually annotate every task-related object for different images under the same task.Compared to previous methods, this approach only requires a single task description as a task-generic prompt for a batch of five annotated samples within the same task, significantly reducing the annotation burden and better aligning with practical needs. Although this setting presents greater challenges than past methods, our ProMaC system achieves excellent performance across a variety of tasks. It even surpasses the performance of state-of-the-art (SOTA) methods trained on weakly supervised datasets for camouflaged object detection, demonstrating the robustness and superior performance of our approach.

## A.2   More Experiments on the Motivation

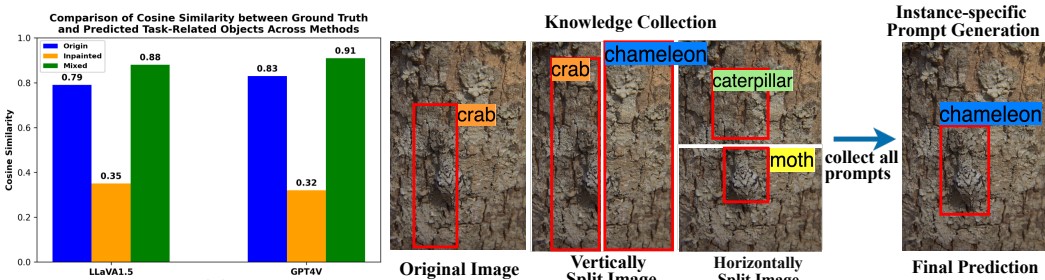

Figure 6: Left: In the bar chart, we analyze MLLM predictions with two versions of an image: the original (blue) and another with task-related objects removed via inpainting (orange). We then compare their predictions to the ground truth using CLIP similarity on the COD10K dataset. Despite missing key objects, the inpainted image's predictions still somewhat match the ground truth. When we select the higher similarity score from both images as the final score (green), it surpassed that of the original alone. It shows that prior knowledge from hallucinations can also provide useful information for generating prompts. Right: A example of using hallucinations to assist instance-specific prompt generation. Specifically, utilizing hallucination can leverage prior knowledge of image elements to better recognize and locate task-related objects. Directly inputting the image into LLaVA results in the hidden chameleon being incorrectly predicted. Splitting the image results in interested objects being incomplete or absent, prompting LLaVA to induce hallucinations and utilize prior knowledge to uncover potential task-related knowledge within the image. This knowledge assists in final accurately identifying and locating the chameleon.

## A.3   Further Demonstration on Implement Details

We conducted experiments across multiple datasets and compared our method with various approaches across different tasks. Initially, we benchmarked against methods such as GPT4V+SAM and LLaVA1.5+SAM to demonstrate that even the current state-of-the-art (SOTA) MLLM methods combined with SAM cannot directly address this issue. Here, GPT4 utilized the gpt-4-vision-preview model, and LLaVA1.5 used the LLaVA-1.5-13B model, with SAM employing the ViT-H/16 version. Both models were tested in a single iteration setup, as our experiments showed that multiple iterations resulted in performance degradation as iterations increased. Specifically, these MLLMs were tasked with inferring instance-specific prompts to guide SAM in segmentation. Similarly, in our setup, we fine-tuned the source code of representative promptable segmentation methods such as SEEM, GroundingSAM, and X-Decoder to adapt them to our setting. These models were directly fed task-generic prompts to segment each image, with performance evaluated based on a single iteration of inference. This methodological adjustment allowed us to assess the efficacy of leveraging task-generic prompts in practical segmentation tasks.

Our ProMaC system employs LLaVA-1.5-13B and the ViT-H/16 version of SAM for inference, utilizing instance-specific prompts generated by the prompt generator. These prompts include instance-specific text prompts and instance-specific bounding boxes. The instance-specific text prompts are processed by Spatial CLIP, mapped to corresponding image regions, and along with the instance-specific bounding boxes, are input into SAM to guide segmentation. All tasks, except for the PIS task, underwent four epochs of iteration, while the PIS task was iterated over six epochs. In the Camouflaged Object Detection task, the task-generic prompt used was "camouflaged animal". We also compared our results with past weakly supervised learning methods, which require at least one point per image in the training set as supervision. Despite requiring less manual supervision with our task-generic promptable segmentation setting, our method achieved comparable or even better performance. Experiments were also conducted in Medical Image Segmentation and Transparent Object Segmentation tasks, with task-generic prompts "polyp" and "skin lesion" respectively for each task. Notably, the datasets used for the transparent object detection task were exclusively glass, allowing the instance-specific text prompt "glass" to be obtained directly without inference, requiring only the inference of bounding boxes through multi-scale chains of thought prompting. This streamlined approach emphasizes the efficiency and adaptability of our system across varied segmentation tasks.

Table 7: Results for Our Method under various task-generic prompt settings.

| Methods | CHAMELEON [50] | | | | CAMO [30] | | | |
|---|---|---|---|---|---|---|---|---|
| | $M \downarrow$ | $F_\beta \uparrow$ | $E_\phi \uparrow$ | $S_\alpha \uparrow$ | $M \downarrow$ | $F_\beta \uparrow$ | $E_\phi \uparrow$ | $S_\alpha \uparrow$ |
| ProMaC | 0.044±0.003 | 0.790±0.003 | 0.899±0.003 | 0.833±0.01 | 0.090±0.002 | 0.725±0.002 | 0.846±0.002 | 0.767±0.002 |

| Methods | COD10K [14] | | | | CVC-ColonDB [51] | | | |
|---|---|---|---|---|---|---|---|---|
| | $M \downarrow$ | $F_\beta \uparrow$ | $E_\phi \uparrow$ | $S_\alpha \uparrow$ | $M \downarrow$ | $F_\beta \uparrow$ | $E_\phi \uparrow$ | $S_\alpha \uparrow$ |
| ProMaC | 0.042± 0.002 | 0.716±0.002 | 0.876±0.003 | 0.805±0.002 | 0.176±0.002 | 0.243±0.002 | 0.583±0.003 | 0.530±0.002 |

| Methods | Kvasir [25] | | | | ISIC [10] | | | |
|---|---|---|---|---|---|---|---|---|
| | $M \downarrow$ | $F_\beta \uparrow$ | $E_\phi \uparrow$ | $S_\alpha \uparrow$ | $M \downarrow$ | $F_\beta \uparrow$ | $E_\phi \uparrow$ | $S_\alpha \uparrow$ |
| ProMaC | 0.166 ± 0.003 | 0.394± 0.002 | 0.726 ± 0.004 | 0.573 ± 0.003 | 0.160 ± 0.003 | 0.729 ± 0.004 | 0.766 ± 0.003 | 0.703 ± 0.004 |

| Methods | GSD [34] | | | | Trans10K-hard [56] | | | |
|---|---|---|---|---|---|---|---|---|
| | $M \downarrow$ | $F_\beta \uparrow$ | $E_\phi \uparrow$ | $S_\alpha \uparrow$ | $M \downarrow$ | $F_\beta \uparrow$ | $E_\phi \uparrow$ | $S_\alpha \uparrow$ |
| ProMaC | 0.147 ± 0.002 | 0.409 ± 0.003 | 0.723± 0.004 | 0.569 ± 0.003 | 0.251 ± 0.003 | 0.509 ± 0.004 | 0.654 ± 0.002 | 0.557 ± 0.002 |

## A.4 Further Experiments Results Analysis and Visualization

In Tab. 7, we present the variance-inclusive experimental results of our ProMaC framework across three major datasets, obtained under three different seeds, with detailed environment setup available in the code instructions provided in the supplementary materials. For the COD dataset, we used LLaVA1.5+SAM as our baseline model to ensure fairness, with the comparative GenSAM model utilizing the same configuration. The three datasets utilized involve challenging scenarios with camouflaged animals or people, where some task-related targets are obscured or very small. Under such complex conditions, ProMaC achieves performance similar to or even better than weakly-supervised training methods with only a brief task description like "camouflaged animal" through test-time adaptation, demonstrating our method's superiority. Similarly, on the MIS and TOD tasks, our method significantly outperforms the comparative promptable segmentation approaches. On the OVS task, we first use LLaVA1.5 to identify which categories from a given multi-class list are actually present in the image. The categories identified are then further reasoned through our ProMaC to derive the final results. What'sUp is a benchmark designed to evaluate the spatial understanding abilities of MLLMs. It comprises 820 images that depict clear spatial relationships between two household items, such as a chair and a bowl, each image exclusively features the two objects in one of four distinct spatial configurations. We regard ProMaC's VCR as a visual marker strategy integrated into LLaVA1.5 to guide spatial reasoning and compared with traditional methods. Compared to previous approaches and baselines, our method significantly enhances performance, also highlighting the efficacy of the VCR module.

Moreover, in Fig.5, we analyze how masks evolve across multiple tasks with iterations. It is evident that as iterations increase, the segmentation results improve, the boundaries of task-related objects become clearer, and some targets initially undetected are progressively recognized. Additionally, in Fig.7-12, we display masks for different tasks and samples of generated inpainting of task-related objects in contrastive samples. These results further demonstrate the versatility and superiority of our method.

**A.5 Limitation**

## A.5 Limitation

We compared ProMaC with the recently introduced GPT-4o+SAM [? 29] approach on the CHAMELEON and CAMO datasets for the COD task, and on the CVC-ColonDB dataset for the MIS task, with results displayed in the table below. It's evident that MLLMs based on GOT-4V [43] and LLaVA1.5 [37] excel in the COD task but experience a significant performance drop in the MIS task. This suggests that LLaVA and GPT4V lack specialized data like polyp detection, which leads to their underperformance in tasks requiring high specificity. This issue, stemming from the generalization limitations of MLLM datasets, is also evident in our ProMaC, based on LLaVA. While ProMaC achieves impressive results on the COD task due to LLaVA's general capabilities, its performance on the more specialized MIS dataset, though better than the baseline, falls short when compared to the COD task. The newly proposed GPT-4o, trained on a broader dataset, outperforms GPT-4V across various fields and shows significant improvement on the MIS task, further highlighting the impact of the underlying MLLM's generalization capabilities on ProMaC. This points to a need for further exploration and research into the generalization potential of foundational MLLM models in future work.

Table 8: Comparison with present SOTA MLLM approaches.

| Methods | Venue | Camouflaged Object Detection | | | | | | | | Polyp Image Segmentation | | | |
| | | CHAMELEON [50] | | | | CAMO [30] | | | | CVC-ColonDB [51] | | | |
| | | $M \downarrow$ | $F_\beta \uparrow$ | $E_\phi \uparrow$ | $S_\alpha \uparrow$ | $M \downarrow$ | $F_\beta \uparrow$ | $E_\phi \uparrow$ | $S_\alpha \uparrow$ | $M \downarrow$ | $F_\beta \uparrow$ | $E_\phi \uparrow$ | $S_\alpha \uparrow$ |
|---|---|---|---|---|---|---|---|---|---|---|---|---|---|
| GPT4V+SAM [43, 29] | Arxiv23 | 0.180 | 0.557 | 0.710 | 0.637 | 0.206 | 0.466 | 0.666 | 0.573 | 0.578 | 0.051 | 0.246 | 0.242 |
| LLaVA1.5+SAM [37, 29] | NeruIPS23 | 0.168 | 0.561 | 0.718 | 0.666 | 0.314 | 0.401 | 0.585 | 0.501 | 0.491 | 0.194 | 0.355 | 0.357 |
| GPT-4o+SAM [? 29] | ArXiv24 | 0.073 | 0.638 | 0.779 | 0.706 | 0.116 | 0.582 | 0.727 | 0.659 | **0.067** | **0.340** | **0.655** | **0.575** |
| ProMaC | Ours | **0.044** | **0.790** | **0.899** | **0.833** | **0.090** | **0.725** | **0.846** | **0.767** | 0.176 | 0.243 | 0.583 | 0.530 |

Table 9: Comparison with present SOTA MLLM approaches.

| Methods | Venue | Polyp Image Segmentation | | | |
| | | CVC-ColonDB [51] | | | |
| | | $M \downarrow$ | $F_\beta \uparrow$ | $E_\phi \uparrow$ | $S_\alpha \uparrow$ |
|---|---|---|---|---|---|
| GPT-4o+SAM [? 29] | ArXiv24 | 0.067 | 0.340 | 0.655 | 0.575 |
| Qwen-14B+SAM | ArXiv24 | 0.189 | 0.271 | 0.533 | 0.536 |
| ProMaC(LLaVA) | Ours | 0.767 | 0.176 | 0.243 | 0.583 |
| ProMaC(Qwen) | Ours | 0.104 | 0.289 | 0.601 | 0.583 |

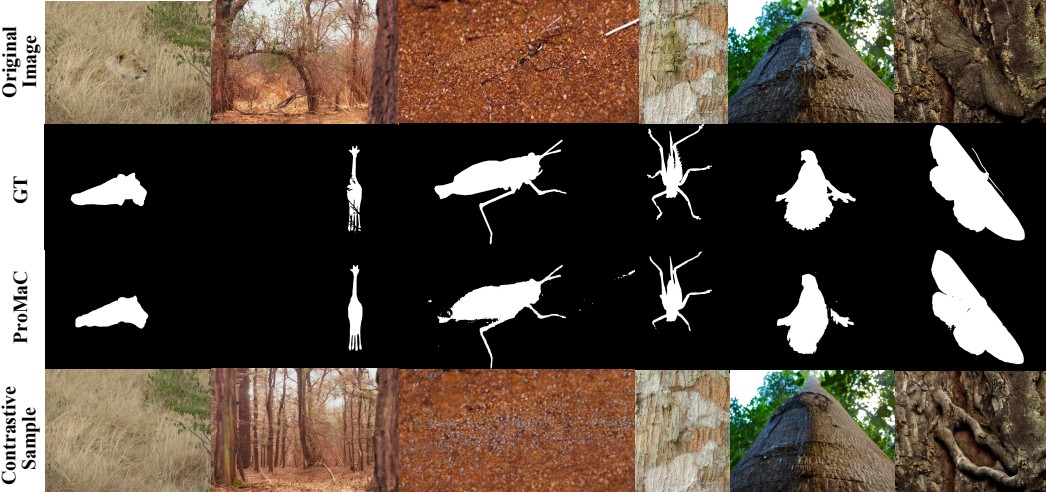

Figure 7: Visualization of the generated mask and contrastive samples on CHAMELEON dataset.

## A.6 Robustness of ProMaC

ProMaC uses MLLM to infer instance-specific prompts that guide the promptable segmentation model. A key question arises: how does the model ensure convergence to the accurate task-related region when initial instance-specific prompts from early iterations are imprecise? To address this, we utilize hallucinations to mine a potential list of instance-specific prompts from multiple scales and repeated image segmentations, ensuring comprehensive exploration of possible task-related objects.

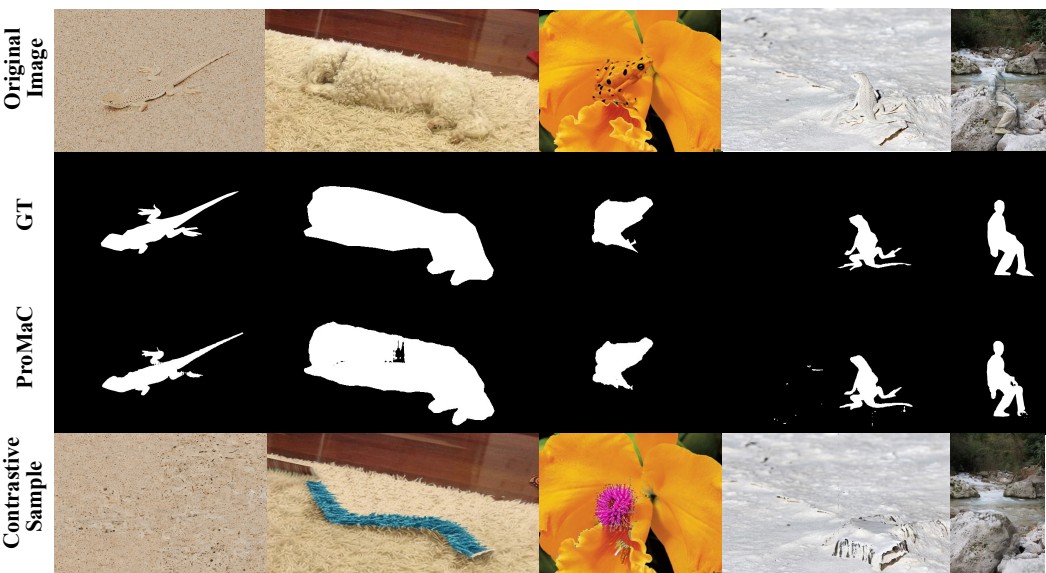

Figure 8: Visualization of the generated masks and contrastive samples on CAMO dataset.

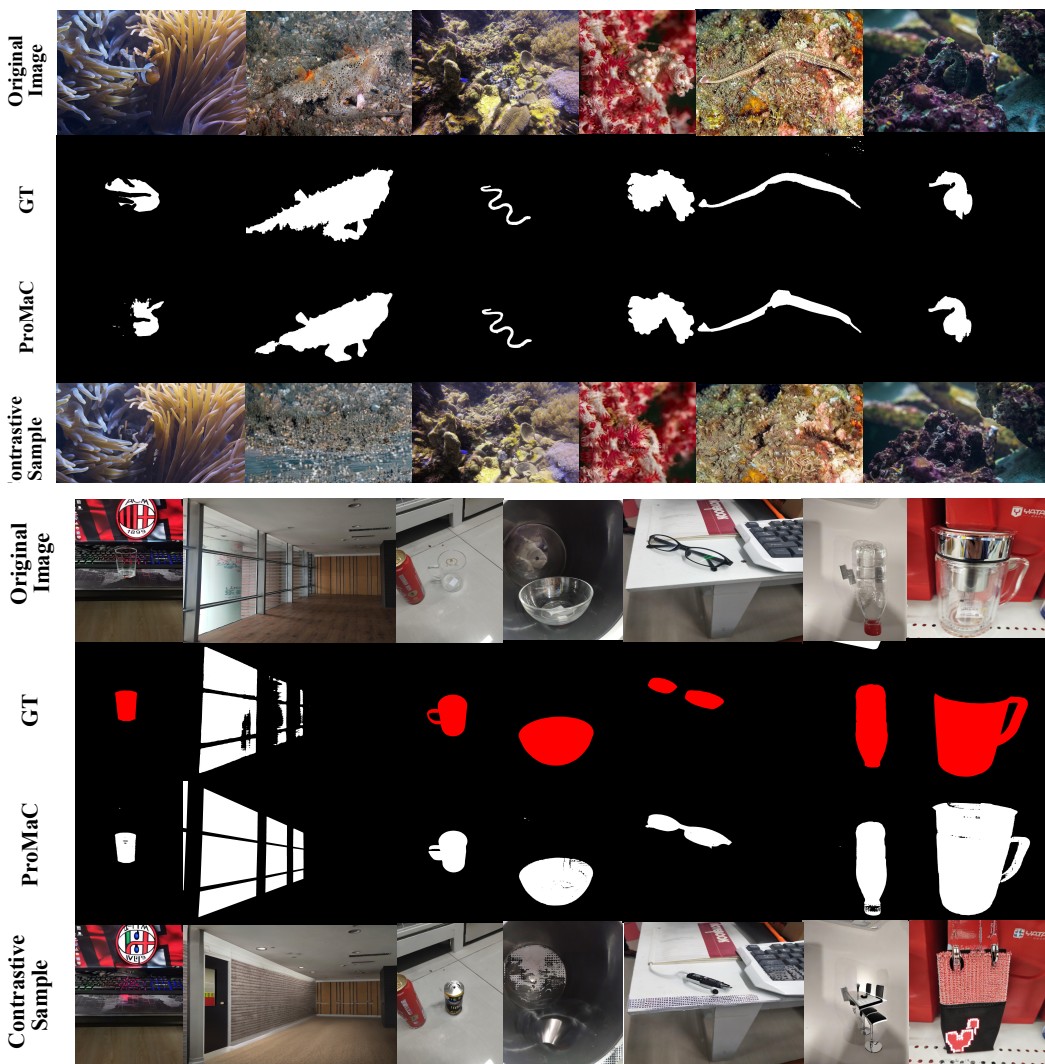

Figure 10: Visualization of the generated masks and contrastive samples on Trans10K dataset.

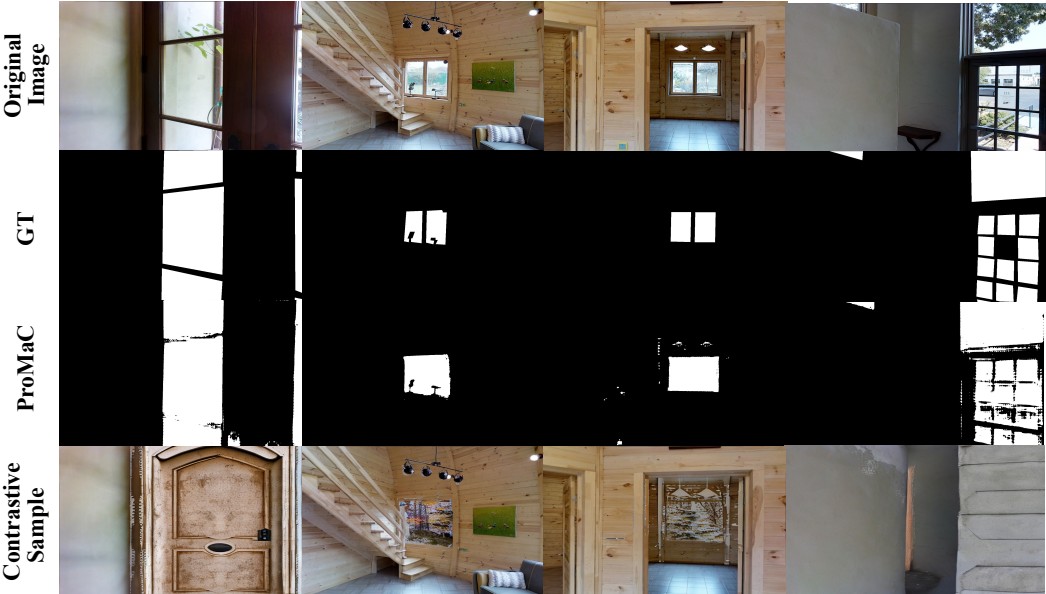

Figure 11: Visualization of the generated masks and contrastive samples on GSD dataset.

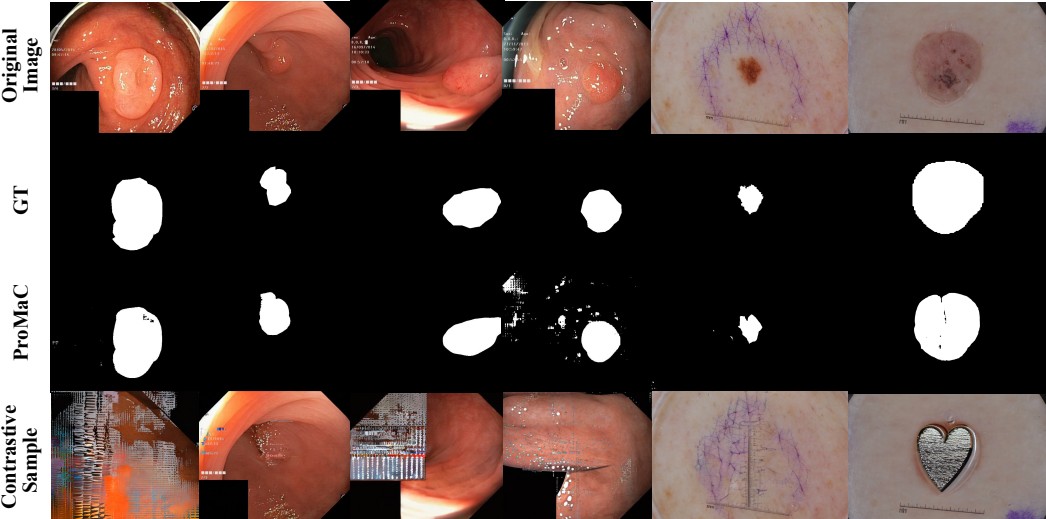

Figure 12: Visualization of the generated masks and contrastive samples on medical dataset.

Subsequently, VCR eliminates less accurate hallucinations to pinpoint the most precise instance-specific prompt from the candidate set. Additionally, our instance-specific prompts include both bounding boxes and keywords, enhancing fault tolerance through a multimodal setup. Experiments in Tab.4 demonstrate the robustness of our multi-modal prompts. Even if the generated prompts do not perfectly match the ground truth, semantic similarity often accurately locates objects of interest within images, facilitating gradual fine-tuning through successive iterations. Even if the initial iteration produces inaccurate prompts, the input for the next iteration remains the original image, preserving correct object information for potential correction in finer-grained iterations. In summary, by providing a wide range of candidates and employing a rigorous strategy for generating instance-specific prompts, along with multimodal prompt inputs and leveraging SAM's generalizability, ProMaC consistently produces accurate instance-specific prompts across various tasks.

### A.7 Social impact

Our work is equivalent to general works in computer vision field, which aims at reducing the manual prompt dependency in promptable segmentation. Therefore, our work has similar potential societal impacts as previous works [21].

---

**Algorithm 1** Algorithm of our ProMaC

---

1: **procedure** PROMAC($X$, LLaVA1.5, SAM, $P_g$, $P_A$, $P_B$)
2:     **Input:** Sample $X \in \mathbb{R}^{H \times W \times 3}$; Multimodal Large Language Model LLaVA1.5, promptable segmentation model SAM, task-generic prompt $P_g$, keyword prompt $P_A$, bbox prompt $P_B$.
3:     **Output:** Segmentation result $M_{i^*}$.
4:     **for** iter $i = 1$ to **I do**
5:         Split image into $K$ multi-scale patches.
6:         **if** $i == 1$ **then**
7:             **for** $k = 1$ to $K$ **do**
8:                 $C^k \leftarrow$ GENERATECAPTION(LLaVA1.5, $X^k$),
9:                 $B^k \leftarrow$ GENERATEBBOX(LLaVA1.5, $X^k$, $C^k$, $P_B$),
10:                $A^k_{\text{fore}}, A^k_{\text{back}} \leftarrow$ GENERATEKEYWORD(LLaVA1.5, $X^k$, $C^k$, $P_A$),
11:             **end for**
12:         **else if** $i! = 1$ **then**
13:             **for** $k = 1$ to $K$ **do**
14:                 $C^k \leftarrow$ GENERATECAPTION(LLaVA1.5, $X^k$),
15:                 $B^k \leftarrow$ GENERATEBBOX(VCR, $X^k$, $X'^k$, $C^k$, $P_B$),
16:                $A^k_{\text{fore}}, A^k_{\text{back}} \leftarrow$ GENERATEKEYWORD(VCR, $X^k$, $X'^k$, $C^k$, $P_A$),
17:             **end for**
18:         **end if**
19:         $\mathcal{A}_i \leftarrow \{A^1_{\text{fore}}, ..., A^K_{\text{fore}}\}, P_A = \{\mathcal{A}_i, P_A\}$,
20:         $\mathcal{B}_i \leftarrow \{B^1, ..., B^K\}, P_B = \{\mathcal{B}_i, P_B\}$,
21:         $X' \leftarrow$ INPAINTIMAGE($X$, $\text{IM}_i$, $P_p$, $P_n$),
22:         $A^u_i, B^u_i \leftarrow$ FINALPROMPT GENERATION($X$, $X'$, $P_A$, $P_B$),
23:         $M_i \leftarrow$ MASKGENERATION($X_i$, $SAM$, $A^u_i$, $B^u_i$),
24:         $X_{i+1} \leftarrow$ WEIGHTEDIMAGE($X_i$, $M_i$),
25:     **end for**
26:     $M_{i^*} \leftarrow$ SELECTBESTMASK($M_i$)
27: **end procedure**

---

