# OpenReview forum: "Leveraging Hallucinations to Reduce Manual Prompt Dependency in Promptable Segmentation"
_NeurIPS.cc/2024/Conference — NeurIPS 2024 poster_

### Official Review · Reviewer_E6it · 2024-07-07

**Soundness:** 3
**Presentation:** 3
**Contribution:** 3
**Rating:** 6
**Confidence:** 5

**Summary:**

The paper introduces an innovative approach called ProMaC (Prompt-Mask Cycle) to improve promptable segmentation. The primary goal of ProMaC is to reduce the dependency on instance-specific manual prompts by leveraging hallucinations from Multimodal Large Language Models (MLLMs) to generate more accurate and task-specific prompts.

**Strengths:**

Innovation in Utilizing Hallucinations: The paper's novel approach of leveraging rather than eliminating hallucinations to improve segmentation tasks is groundbreaking. It recognizes the potential of hallucinations to provide contextual insights, which can be valuable for enhancing model performance.

Reduction of Manual Effort: By introducing task-generic prompts and iteratively refining them, ProMaC significantly reduces the need for manual annotation, making it more feasible for large-scale applications.

Iterative Refinement Process: The iterative approach of ProMaC, which continuously improves prompts and masks, ensures higher accuracy and adaptability across various tasks and datasets.

Comprehensive Evaluation: The paper provides extensive evaluations on multiple benchmarks, demonstrating the robustness and effectiveness of ProMaC in diverse and challenging scenarios.

Adaptability and Versatility: ProMaC's success across different segmentation tasks, including those in medical imaging and transparent object detection, highlights its versatility and potential for broad application.

Open-Source Contribution: The inclusion of code in the supplemental materials encourages further research and development, facilitating community engagement and collaboration.

In summary, the paper makes a significant contribution to the field of promptable segmentation by introducing a novel method that leverages MLLM hallucinations, reduces manual dependency, and demonstrates superior performance across various challenging tasks.

**Weaknesses:**

Reliance on Hallucinations:

Inconsistent Performance: The approach relies on hallucinations, which can be unpredictable and inconsistent. In scenarios where hallucinations are highly inaccurate, the performance of ProMaC could degrade significantly.
Dependence on MLLM Quality: The effectiveness of leveraging hallucinations is heavily dependent on the quality and training of the Multimodal Large Language Models (MLLMs). Variations in the MLLM's training data and methodology can lead to differing hallucination patterns, affecting ProMaC's reliability.
Complexity of the Iterative Process:

Computational Overhead: The iterative process of refining prompts and masks can be computationally intensive. This complexity might limit the scalability of ProMaC for real-time applications or large datasets.
Implementation Challenges: The method's implementation is intricate, involving multiple stages of hallucination generation, contrastive reasoning, and mask refinement. This complexity can pose challenges for practitioners looking to adopt the method.
Generalization Concerns:

Task-Specific Fine-Tuning: While ProMaC aims to reduce manual prompt dependency, the initial setup still requires careful selection of task-generic prompts. Ensuring that these prompts are effective across diverse tasks without additional fine-tuning could be challenging.
Transferability: The approach might not generalize well to entirely new tasks or domains where the hallucinations generated by MLLMs do not align well with the actual task requirements.

**Questions:**

see the weakness

**Limitations:**

see the weakness

---

> ### Author Rebuttal · Authors · 2024-08-06
>
> We thank the reviewer for valuable feedback and for appreciating the idea of our method, strong justification of each component of our framework and extensiveness of our experiments. Following are the responses regarding your concerns.
>
> > *Inconsistent Performance: In scenarios where hallucinations are highly inaccurate, the performance of ProMaC could degrade significantly.*
>
> - Hallucinations are used solely to gather task-relevant information from the image and are effectively filtered by the Visual Contrastive Reasoning module and the subsequent Mask Semantic Alignment model to eliminate the negative impact of task-irrelevant information. This ensures the generated instance-specific prompts are accurate and not influenced by irrelevant hallucinations, maintaining robust performance even when hallucinations are unreliable.
>
> - To elaborate, as stated in Lines 143-150, by dividing the input image into patches at various scales, MLLM induces hallucinations based on varying object visibility. These hallucinations use prior knowledge to explore connections between the image data and the associated task, gathering task-relevant information. Simultaneously, the original image is also fed directly into the MLLM to obtain task-relevant candidates without relying on hallucinations. This ensures that even when hallucinations are highly inaccurate, the collected information still includes relatively reliable content.
>
> - Visual Contrastive Reasoning module then selectively reduces irrelevant semantics to remove irrelevant influences while benefiting from relevant insights obtained from pre-trained knowledge (hallucinations). When hallucinations are completely inaccurate, the information derived from them is discarded. Additionally, the Mask Semantic Alignment module further ensures that the generated masks align with the task semantics.
>
> > *Dependence on MLLM Quality.*
>
> As we explained in the limitations section, the quality of MLLMs affects our model's generalization ability, which is a potential direction for our future research.
>
> > *Computational overhead and implementation challenges.*
>
> - Computational Overhead: Traditional methods often require multiple GPUs for extensive training, leading to higher computational and memory demands. In contrast, as we discussed in Line 281, our method performs iterative test-time adaptation on a single 40G A100 GPU without the need for training, which is computationally efficient. Compared to GenSAM [19], which also uses test-time adaptation, our method achieves better performance with fewer iterations. As shown in Tab. 6(a), ProMaC reaches higher performance by the second iteration than GenSAM does after six iterations, further demonstrating the efficiency of our approach.
>
> - Implementation Challenges: Although our approach uses multiple modules, all module parameters are fixed and do not require training, making it easier to reproduce. In Tab. 7, we conducted three sets of experiments randomly and calculated the mean and variance. The results show that our training-free method exhibits very low variance, further proving its robustness. Additionally, we provide implementation details in the PyTorch Implementation Details section (Line 271-281) of the main text and Line 551-581 of the appendix. Our code is also available in the supplementary materials, ensuring that our method is easy to understand and implement.
>
> > *The initial setup still requires careful selection of task-generic prompts. Ensuring that these prompts are effective across diverse tasks without additional fine-tuning could be challenging.*
>
> - Task-generic prompt $P_g$ is decided by the task name rather than our manual selection, and has minimal impact on experimental results. Using the camouflaged animal detection task as an example, we selected "camouflaged animal" as $P_g$ because the task is named "camouflaged animal detection". Similarly, "polyp" is selected as $P_g$ because the corresponding task is named as "polyp detection".
> - To explore the impact of different $P_g$ prompts, we used ChatGPT to suggest two synonyms for "camouflaged animal": "hidden animal" and "disguised animal" as possible $P_g$ candidates. We evaluated the effect of different $P_g$, as shown in Tab. 2 of the attached PDF file. Although different $P_g$ prompts cause slight fluctuations in the outcomes, the overall performance remains stable. This demonstrates that as long as $P_g$ effectively describes the task semantics, our method can achieve relatively consistent segmentation performance.
>
> > *Transferability: The approach might not generalize well to entirely new tasks or domains where the hallucinations generated by MLLMs do not align well with the actual task requirements.*
>
> - Because our ProMaC can be easily applied to different MLLMs, we can ensure robust performance across various tasks and domains by selecting the most appropriate MLLM for each specific task.
>
> - PoMaC is designed to be flexible and can be adapted to different MLLMs that are more suitable for specific tasks or domains. By selecting an MLLM that has been pre-trained on data relevant to the target domain, we can improve alignment with task requirements and enhance performance.
>
> - For example, as shown in Tab. 1 in attached pdf, in medical imaging segmentation, ProMaC based on LLaVA performed modestly. However, when we used LLaVA-Med [1], which has stronger medical characteristics, as the base, ProMaC significantly outperformed both LLaVA-Med combined with SAM and LLaVA-based ProMaC. This demonstrates that by applying our ProMaC method to a more suitable base MLLM, we can achieve better performance, thereby ensuring transferability across various tasks.
>
> [1] Li C, Wong C, Zhang S, et al. Llava-med: Training a large language-and-vision assistant for biomedicine in one day[J]. Advances in Neural Information Processing Systems, 2024, 36.

---

> > ### Comment · Reviewer_E6it · 2024-08-08
> >
> > thanks the rebuttal from the authors. they solved all my concerns. I agreed with all the other reviewers that we should accept the paper.

---

### Official Review · Reviewer_u8FA · 2024-07-08

**Soundness:** 3
**Presentation:** 2
**Contribution:** 3
**Rating:** 6
**Confidence:** 4

**Summary:**

The paper focuses on promptable segmentation, aiming to minimize the need for manually designing prompts. Specifically, it explores the hallucination issue in MLLMs and finds that hallucinations can reveal valuable contextual information, which could be largely beneficial to promptable segmentation tasks, especially for complex tasks such as camouflaged object segmentation. To this end, the paper proposes a Prompt-Mask Cycle generation framework (ProMaC) that iteratively refines the generated prompts and masks. The proposed method leverages hallucinations to mine more context-related information while simultaneously reducing irrelevant hallucinations for better segmentation results.

**Strengths:**

1. The idea of this paper is quite interesting. Instead of merely mitigating hallucinations, the paper explores how to leverage them for better segmentation performance. The paper also provides detailed analysis to explain its motivation and illustrate how hallucinations can benefit the segmentation task.

2. Extensive experiments over diverse segmentation benchmarks demonstrate the effectiveness of the proposed method in tackling various segmentation tasks, including very challenging camouflage segmentation and general segmentation tasks.

**Weaknesses:**

Motivation

1. The author illustrates the motivation mainly using samples from the camouflaged object segmentation task (in Figure 1 and A.2). This makes it quite straightforward to understand the rationale behind the proposed method and how it can benefit the camouflaged object segmentation task. However, it would be better to involve more examples from different segmentation tasks to illustrate the motivation comprehensively, especially general segmentation like COCO and Pascal VOC. I am curious whether this phenomenon would still occur when the objects are more visible


Method

1. Some terms are unclear and need explanation. For example, I’m not very clear about the meaning of the term ‘prompt’ used in the paper. Does it refer to text prompts or spatial prompts like boxes or points? From the method section, it seems that P_B and P_A are text prompts, and the instance-specific prompts refer to bounding boxes? Also, what does query P refer to? If its meaning differs from P_g, P_B, and P_A, it would be better to clarify this at the beginning of the method section. Since the method is based on prior work [19] and some terms are borrowed directly from [19], it is necessary to include a preliminary section to explain the background and terms. This would make the method section easier to follow.


2. Question about the proposed Multi-scale Chain of Thought Prompting. From my understanding, chain-of-thought prompting aims to achieve complex reasoning capabilities through multiple intermediate reasoning steps. Could the author please explain how this idea is reflected in the proposed method?

**Questions:**

Overall, the idea of this paper is quite interesting. However, it lacks some necessary explanations about the motivation and the proposed method. For example, the motivation is primarily illustrated using examples from camouflaged object segmentation. More diverse examples could help illustrate its broader potential. Additionally, some terms are unclear, suggesting a need for clearer definitions and possibly a background section explaining the foundational concepts.

**Limitations:**

The author has discussed the limitations of the work.

---

> ### Author Rebuttal · Authors · 2024-08-06
>
> We thank the reviewer for the positive evaluation of our work and for acknowledging the originality of the proposed method as well as its significance for future research. We are glad that the reviewer finds the results impressive and the idea of this paper is quite interesting.
>
> > *It would be better to involve more examples from different segmentation tasks to illustrate the motivation comprehensively, especially general segmentation like COCO and Pascal VOC. I am curious whether this phenomenon would still occur when the objects are more visible.*
>
> We have also included additional samples for the polyp dataset and general object dataset in the attached PDF (Fig. 1) to illustrate our motivation. It is evident that our method, although specifically designed for segmentation tasks where visual cues are weak or ambiguous (e.g., hidden/camouflaged foreground objects in visually similar backgrounds), can also utilize hallucinations as prior knowledge to identify potential categories in complex images in general task, such as those in the VOC dataset (Fig. 1(b) in the attached PDF). However, when visual cues are strong, obvious, or distinct, the benefit of scene understanding through exploring hallucinations becomes less critical. In such situations, there is less likelihood of confusion between foreground and background, and less need for reasoning about all plausible backgrounds versus foregrounds, in such cases, our method may not provide significant improvements. (Fig. 1(c) in the PDF).
>
> > *Some terms are unclear and need explanation. For example, I’m not very clear about the meaning of the term ‘prompt’ used in the paper. Does it refer to text prompts or spatial prompts like boxes or points? From the method section, it seems that P_B and P_A are text prompts, and the instance-specific prompts refer to bounding boxes? Also, what does query P refer to? If its meaning differs from P_g, P_B, and P_A, it would be better to clarify this at the beginning of the method section. Since the method is based on prior work [19] and some terms are borrowed directly from [19], it is necessary to include a preliminary section to explain the background and terms. This would make the method section easier to follow.*
>
> The term 'prompt' used in the paper includes both text prompts and spatial prompts (bounding boxes and points).
>
> - As explained in Line 152-155, in each task, the input text prompt consists of two parts: prompt for bounding box prediction ($P_B$) and prompt for class prediction ($P_A$):
>
> 1. Prompt for bounding box prediction ($P_B$) instructs: "This image is from the $P_g$ detection task, output the bounding box of the $P_g$."
>
> 2. Prompt for class prediction ($P_A$) states: "Output the name of the $P_g$ and its environment in one word."
>
> Here, $P_g$ is a test-generic prompt that is consistent across tasks. For example, for camouflaged animal detection task, $P_g$ is "camouflaged animal", for polyp detection task, $P_g$ is "polyp".
> - $P_B$ and $P_A$ are fed into MLLM to infer instance-specific spatial prompt bounding box and instance-specific text prompt class name. This class name is then mapped into anther instance-specific spatial point prompts using spatial CLIP. These instance-specific spatial prompts (both point prompts and bounding box prompts) are subsequently fed into SAM to guide the segmentation process.
>
> - We refer to the queries inputted into the MLLM as P.
> - The terms borrowed from [19] will be explained in more detail in the final version.
>
> > *Question about the proposed Multi-scale Chain of Thought Prompting. From my understanding, chain-of-thought prompting aims to achieve complex reasoning capabilities through multiple intermediate reasoning steps. Could the author please explain how this idea is reflected in the proposed method?*
>
> As we explained in Line 143-190, our MCoT uses multiple intermediate reasoning steps on various chains to infer accurate instance-specific prompts from task-generic prompts.
> - First, as we mentioned in Line 146-156, each chain includes two intermediate reasoning steps: 1. MLLM first captions the image and 2. Based on this captions, MLLM infers the names and backgrounds of task-relevant objects. Without this initial image captioning step, the inferred instance-specific prompts would be inaccurate.
> - Second, The input image scales differ across various chains, leading to variations in the positions and completeness of task-relevant objects in each chain. This difference results in distinct intermediate reasoning paths, ensuring diverse information extraction across chains. This variation allows the MLLM to fully leverage prior knowledge from hallucinations to uncover potential task-relevant information.
> - Finally, the visual contrastive reasoning module aggregates effective information from different chains, eliminating the influence of task-irrelevant hallucinations. This multi-chain, multi-step reasoning process ultimately produces accurate instance-specific prompts.

---

> > ### Comment · Reviewer_u8FA · 2024-08-10
> >
> > Thank the authors for the response. All my concerns have been well addressed. I'd like to raise my score to weak accept.

---

### Official Review · Reviewer_fu4N · 2024-07-09

**Soundness:** 4
**Presentation:** 3
**Contribution:** 4
**Rating:** 7
**Confidence:** 5

**Summary:**

The paper introduces the Prompt-Mask Cycle generation framework (ProMaC), which innovatively uses hallucinations from Multimodal Large Language Models to refine segmentation prompts and masks. This method contrasts with traditional approaches by leveraging rather than eliminating hallucinations, enhancing task-related accuracy through an iterative prompting and masking process. ProMaC's efficacy is demonstrated across various benchmark.

**Strengths:**

- Unlike previous methods that considered hallucinations as negative, the authors view them as prior knowledge from the pre-trained model, first extracting task-relevant information and then validating it. This perspective is very insightful.
- The article is well-structured and clearly written, making it easy to follow.
- The experiments are comprehensive, conducted across various tasks and diverse datasets, demonstrating the method's effectiveness and robustness. Ablation studies are also thoroughly conducted on datasets from different tasks to better illustrate the contribution of each component.
- The idea of combining SAM and inpainting to generate images without task-related objects to eliminate hallucinations is intriguing and could be adapted to more works aiming to reduce hallucinations.

**Weaknesses:**

- The authors demonstrate the outstanding performance of the proposed ProMaC method across various tasks. Could they provide comparison results between the predictions of the ProMaC method and the ground truth across iterations? This would offer a more visual representation of the method's performance.
- The proposed method's computational and memory requirements aren't clearly discussed, which might be significantly higher due to the iterative nature of the Prompt-Mask Cycle and multiscale chain of thought prompting. This could limit its applicability in resource-constrained environments.

**Questions:**

1)Dependence on Initial Prompts: While the iterative refinement process is beneficial, the initial quality of prompts still plays a crucial role. Poor initial prompts might lead to suboptimal starting points, impacting the overall effectiveness of the adaptation process. I would like to see some discussion on this.
2)The authors utilize the LLaVA1.5 as the base MLLM for experiments, achieving results comparable to weakly-supervised and even supervised training in tasks such as Camouflaged Animal Detection, whereas performance in tasks like Medical Image Segmentation is more modest. The authors attribute this to the inherent generalization limitations of LLaVA. Recently developed methods (e.g., GPT4o) have shown better generalization capabilities in medical imaging. Could the authors conduct experiments using these more robust MLLMs on specific sub-tasks to substantiate this perspective?

**Limitations:**

The paper encounters a few limitations, such as the potential dependence on the initial quality of prompts and potential resource intensity due to the iterative and computationally demanding nature of the ProMaC method. Additionally, the performance disparity across different tasks highlights the limitations of the underlying MLLM's generalization capabilities. Despite these challenges, the overall contribution of the paper remains significant. The novel approach to leveraging hallucinations as a resource rather than a drawback, coupled with comprehensive experiments and detailed reproducibility, underscores the paper's value.

---

> ### Author Rebuttal · Authors · 2024-08-06
>
> We thank the reviewer for the positive evaluation of our work and for acknowledging the value and insight of the proposed method.
>
> > *Could they provide comparison results between the predictions of the ProMaC method and the ground truth across iterations? This would offer a more visual representation of the method's performance.*
>
> On Page 16, in Fig. 6, we provide visualizations of segmentation results across different tasks as iterations progress. These results demonstrate that our method shows consistent performance improvement with each iteration across various tasks. Additionally, the quantitative experimental results in Tab. 6 further corroborate this finding, illustrating the incremental enhancement in performance as the iterations proceed.
>
> > *The proposed method's computational and memory requirements aren't clearly discussed, which might be significantly higher due to the iterative nature of the Prompt-Mask Cycle and multiscale chain of thought prompting. This could limit its applicability in resource-constrained environments.*
>
> Traditional methods often require multiple GPUs for extensive training, leading to higher computational and memory demands. In contrast, as we discussed in Line 281, our method performs iterative test-time adaptation on a single 40G A100 GPU without the need for training, which is computationally efficient. Compared to GenSAM, which also uses test-time adaptation, our method achieves better performance with fewer iterations. As shown in Tab. 6(a), ProMaC reaches higher performance by the second iteration than GenSAM does after six iterations, further demonstrating the efficiency of our approach.
>
> > *Dependence on Initial Prompts: While the iterative refinement process is beneficial, the initial quality of prompts still plays a crucial role. Poor initial prompts might lead to suboptimal starting points, impacting the overall effectiveness of the adaptation process. I would like to see some discussion on this.*
>
> The format of the initial prompt is fixed, with the only modifiable part being the task-generic prompt $P_g$. However, $P_g$ has minimal impact on experimental results. Using the camouflaged animal detection task as an example, we selected "camouflaged animal" as $P_g$ because the task is named "camouflaged animal detection". To explore the impact of different $P_g$ prompts, we used ChatGPT to suggest two synonyms for "camouflaged animal": "hidden animal" and "disguised animal" as possible $P_g$ candidates. We evaluated the effect of different $P_g$ on the results, as shown in Tab. 2 of the attached PDF file. Although different $P_g$ prompts cause slight fluctuations in the outcomes, the overall performance remains stable. This demonstrates that as long as $P_g$ effectively describes the task semantics, our method can achieve relatively consistent segmentation performance.
>
> >*The authors attribute this to the inherent generalization limitations of LLaVA. Recently developed methods (e.g., GPT4o) have shown better generalization capabilities in medical imaging. Could the authors conduct experiments using these more robust MLLMs on specific sub-tasks to substantiate this perspective?*
>
> In Tab. 8 on Page 16, we compare the performance of our method with different baseline methods using various MLLMs across different tasks. It is evident from the comparisons among the different baselines that the generalization capability of an MLLM directly impacts its performance on different tasks, which supports our explain. Furthermore, we evaluate an MLLM model based on LLaVA-Med [1] on the Polyp Image Segmentation task, as shown in Tab.1 in attached pdf file, which also support this claim.
>
> [1] Li C, Wong C, Zhang S, et al. Llava-med: Training a large language-and-vision assistant for biomedicine in one day[J]. Advances in Neural Information Processing Systems, 2024, 36.

---

> > ### Comment · Reviewer_fu4N · 2024-08-13
> >
> > The authors addressed all my concerns. I suggest to accept this paper.

---

### Official Review · Reviewer_E5Cz · 2024-07-11

**Soundness:** 3
**Presentation:** 3
**Contribution:** 3
**Rating:** 6
**Confidence:** 4

**Summary:**

The paper introduces an iterative Prompt-Mask Cycle generation framework (ProMaC) with a prompt generator and a mask generator. The prompt generator uses a multi-scale chain of thought prompting, initially exploring hallucinations for extracting extended contextual knowledge on a test image. These hallucinations are then reduced to formulate precise instance-specific prompts, directing the mask generator to produce masks that are consistent with task semantics by mask semantic alignment. The generated masks iteratively induce the prompt generator to focus more on task-relevant image areas and reduce irrelevant hallucinations, resulting jointly in better prompts and masks. The results on several challenging datasets validate the effectiveness of it.

**Strengths:**

-A task-generic prompt is used to prompt the proposed ProMaC to perform segmentation. The results on several challenging datasets validate the effectiveness of it.

-The authors use useful techniques, multi-scale chain of Thought Prompting, visual Contrastive Reasoning, and Contrastive Sampling Generation, to generate accurate instance-specific prompts.

**Weaknesses:**

-The authors mentions that the main motivation of this paper is to utilize hallucinations instead of eliminate it. However, the VCR module is used to “reducing” and “minimizing” it. This confuses me. The authors should clearly explain how this paper utilizes the hallucinations.

- The iterative process and the integration of various techniques (e.g., MLLM, SAM, CLIP, and inpainting) add complexity, which may hinder the ease of implementation and understanding.

-Compared with GenSAM [19]， What is the main innovation of the Mask Generator?

- At the first iteration, since masks have not yet been generated, what are the visual markers in this process?

-The core of this task is how to generate accurate visual prompts through MLLM using a general text prompt. From my perspective, this task should be evaluated more on general datasets, such as COCO, to demonstrate the method's generalizability rather than on specific domain tasks (e.g., camouflage). Some comparison methods in this paper, such as X-Decoder and SEEM, are all generalized methods. Comparing with them, I believe, is unreasonable. I hope the author can discuss this issue with me.

-Minor: The authors should list the genetic task prompts used for different tasks.

**Questions:**

See the weakness part.

**Limitations:**

The paper presents the limitation in the appendix. The authors suggest further exploration and research into the generalization potential of foundational MLLM models in future work.

---

> ### Author Rebuttal · Authors · 2024-08-06
>
> We thank the reviewer for the careful reading and insightful comments. Following are the responses regarding your concerns.
>
> > *The authors should clearly explain how this paper utilizes the hallucinations.*
>
> As we explained in Line 143-150, we utilize hallucination to bootstrap a scene-understanding of each test image. By dividing the input image into patches at various scales, the MLLM induces hallucinations based on varying object visibility in diverse patches. These hallucinations use prior knowledge to explore connections between the image data and the associated task, gathering task-relevant information before eliminating irrelevant semantics in our reasoning process. This is critical when visual cue is weak / ambiguous for segmentation. We then selectively reduce irrelevant semantics in order to remove irrelevant influence whilst benefit from relevant influence obtained from pre-trained knowledge (aka hallucination). This is in contrast to existing methods which all aim to remove hallucination blindly regardless its usefulness in helping scene-understanding in an image, therefore improving segmentation when visual cues are weak / ambiguous.
>
> >  *The iterative process and the integration of various techniques (e.g., MLLM, SAM, CLIP, and inpainting) add complexity, which may hinder the ease of implementation and understanding.*
>
> - Traditional methods often require multiple GPUs for extensive training, leading to higher computational and memory demands. In contrast, as we discussed in Line 281, our method performs iterative test-time adaptation on a single 40G A100 GPU training-free, meaning all module parameters are fixed from the start. This eliminates the need for training and parameter tuning, and therefore reducing implementation complexity.
>
> - We also provide implementation details in the PyTorch Implementation Details section (Line 271-281) of the main text and Line 551-581 of the appendix. Our code is available in the supplementary materials, ensuring our method is easy to understand and implement.
>
> > *Compared with GenSAM [19]， What is the main innovation of the Mask Generator?*
>
> - As we explained in Line 212-213, the key innovation of our mask generator is the proposed Mask Semantic Alignment (MSA) module, which ensures that the generated masks align with task semantics—an achievement that GenSAM cannot match.
>
> - In GenSAM, the mask generator uses spatial CLIP to convert instance-specific text prompts into positive/negative point prompts and directly feeds them into SAM to obtain masks. However, since SAM is trained without category labels, it lacks label prediction capabilities. This leads to masks that do not meet task requirements when the point prompts fed into SAM are not accurate.
>
> - Our MSA module addresses this limitation by using similarity-weighted masks aggregation to ensure that the output masks are aligned with the task semantics. The effectiveness of our MSA module is demonstrated in the last two rows of the ablation experiment in Tab. 4 in original paper.
>
> > *At the first iteration, since masks have not yet been generated, what are the visual markers in this process?*
>
> As we explained in Line 199-201, in the first iteration, Multi-scale Chain of Thought Prompting (MCoT) infers predicted bounding boxes for multiple different patches. We use the union of these bounding boxes as the visual marker for the first iteration to guide the generation of the contrastive sample $X^{'}$.
>
> > *From my perspective, this task should be evaluated more on general datasets, such as COCO, to demonstrate the method's generalizability rather than on specific domain tasks (e.g., camouflage). Some comparison methods in this paper, such as X-Decoder and SEEM, are all generalized methods. Comparing with them, I believe, is unreasonable. I hope the author can discuss this issue with me.*
>
> - In Tab. 3(b), we have conducted experiments on three general datasets: COCO Object, PASCAL VOC, and Pascal Context, showing that our method achieves good performance on general segmentation tasks comparing to other unsupervised/weakly supervised methods.
>
> - As for generalized methods like X-Decoder and SEEM, it is ideal to fairly compare X-Decoder and SEEM on both general and specific tasks. However, these two methods are trained on general segmentation datasets with semantic labels, while our pre-trained model only trains with image mask pairs without semantic labels. Therefore, it is unfair to directly compare ProMaC with these methods on general segmentation tasks. Meanwhile, since neither these methods nor our ProMaC have been trained on specific domain datasets, comparing them on these tasks is fair.
>
> > *List the genetic task prompts used for different tasks.*
>
> As we explained in Line 273-276, the text generic prompt $P_g$ for the camouflaged animal detection task is “camouflaged animal”. For the two sub-tasks in medical imaging, the text generic prompts are “polyp” for polyp image segmentation and “skin lesion” for skin lesion segmentation. For the transparent object detection task, the text generic prompt is “glass”.

---

> > ### Comment · Reviewer_E5Cz · 2024-08-13
> >
> > After reading the author's response, most of my concerns have been addressed. I've decided to raise my current score.

---

### Author Rebuttal · Authors · 2024-08-06

We thank all the reviewers for their uniformly positive evaluations and valuable feedback. We appreciate fruitful suggestions of the reviewers that helped to improve the overall presentation of our work. We are encouraged by the positive comments from reviewers for the following: (i) The idea of this paper is quite interesting (Reviewers u8FA, fu4N, E6it, E5Cz) (ii) extensive experiments over diverse segmentation benchmarks demonstrate the effectiveness of the proposed method (Reviewers u8FA, fu4N, E6it) and (iii) the writing quality and the clarity of the presentation of the ideas (Reviewer fu4N). Additionally, as mentioned by reviewer E6it, we emphasize that another major strength of our algorithm is that it can reduction of manual effort and have good adaptability and versatility.

In the response to the reviewers’ feedback, we have conducted additional experiments and motivation visualizations shown in the attached pdf. The attached pdf includes:

- Fig. 1: More motivation visualization on medical image and general image tasks.
- Tab. 1: ProMaC performance on the Polyp Image Segmentation task using LLaVA-Med.
- Tab. 2: Performance comparison of ProMaC with different task-generic prompts $P_g$.

We provide more details and address reviewers’ comments in the individual response to reviewers. We hope that our detailed response and additional experiments will help to increase reviewers’ confidence.

---

### Decision · Program_Chairs · 2024-09-25

**Decision:**

Accept (poster)

**Comment:**

Paper was reviewed by four expert reviewers and, post-rebuttal, received uniformly positive scores; mainly: 3 x Weak Accepts and 1 x Accept. While prior to rebuttal reviewers raised some concerns with the work, those concerns have been resolved in rebuttal and discussion. Specifically, [fu4N, E6it, u8FA] acknowledge that "all concerns have been well addressed", while [E5Cz] mentions that "most of  concerns have been addressed".  Overall, all reviewers agree that the approach is novel, interesting and the paper is well written. AC agrees with the reviewers and believes that the paper will be a welcome addition to NeurIPS. Therefore the decision is to Accept the paper. Authors are encouraged to incorporate elements of the rebuttal and discussion into the paper for the camera ready.